# Towards Macroporous α-Al_2_O_3_—Routes, Possibilities and Limitations

**DOI:** 10.3390/ma13071787

**Published:** 2020-04-10

**Authors:** Simon Carstens, Ralf Meyer, Dirk Enke

**Affiliations:** 1Universität Leipzig, Institute of Chemical Technology, Linnéstr. 3, D-04103 Leipzig, Germany; 2Universität Osnabrück, Institute of Chemistry of New Materials, Barbarastr. 7, D-49076 Osnabrück, Germany

**Keywords:** high surface area α-alumina, macroporosity, calculation of specific surface areas, sol-gel, AAO membranes, solid solutions of Mn in alumina, pore protection

## Abstract

This article combines a systematic literature review on the fabrication of macroporous α-Al_2_O_3_ with increased specific surface area with recent results from our group. Publications claiming the fabrication of α-Al_2_O_3_ with high specific surface areas (HSSA) are comprehensively assessed and critically reviewed. An account of all major routes towards HSSA α-Al_2_O_3_ is given, including hydrothermal methods, pore protection approaches, dopants, anodically oxidized alumina membranes, and sol-gel syntheses. Furthermore, limitations of these routes are disclosed, as thermodynamic calculations suggest that γ-Al_2_O_3_ may be the more stable alumina modification for *A_BET_* > 175 m^2^/g. In fact, the highest specific surface area unobjectionably reported to date for α-Al_2_O_3_ amounts to 16–24 m^2^/g and was attained via a sol-gel process. In a second part, we report on some of our own results, including a novel sol-gel synthesis, designated as *mutual cross-hydrolysis*. Besides, the Mn-assisted α-transition appears to be a promising approach for some alumina materials, whereas pore protection by carbon filling kinetically inhibits the formation of α-Al_2_O_3_ seeds. These experimental results are substantiated by attempts to theoretically calculate and predict the specific surface areas of both porous materials and nanopowders.

## 1. Preface

Aluminum oxide Al_2_O_3_ exists in a variety of modifications [1]. The so-called transition alumina, such as γ-, κ-, η-, δ-, or θ-Al_2_O_3_, are all easily moldable and exhibit a certain porosity, rendering these modifications suitable for various applications, for instance, in the field of heterogeneous catalysis. Corundum, as α-Al_2_O_3_ is commonly designated, is the thermodynamically stable modification, for *p* < 400 GPa [2,3,4]. However, it is intrinsically non-porous and hence inapt for any catalytical application, apart from being used as a support material [5]. Introducing porosity into α-Al_2_O_3_ materials is a major challenge, having given rise to a considerable quantity of scientific and patent literature. Apart from representing a major scientific and technological progress, porous α-Al_2_O_3_ could be employed in different highly relevant fields of application, including catalytic cracking and steam reforming [6], filtration of hot gases [7], and ultrafiltration membranes [8].

This article provides a comprehensive literature review of attempts towards high specific surface area or porous α-Al_2_O_3_, which by material constraints will always be macroporous. The authors believe to have gathered and assessed information on all major routes for the preparation of porous α-Al_2_O_3_ published to date.

In a second part, calculations of specific surface areas for both porous materials and nanopowder are suggested and applied to results from our group. Lastly, recent results from our laboratory are presented, showing both new possibilities to obtain α-Al_2_O_3_ with enhanced porosity and limitations of certain routes.

## 2. Literature Review: Synthesis Routes for α-Al_2_O_3_ with Increased Specific Surface Area

### 2.1. Introduction

Two main pathways lead to α-Al_2_O_3_, starting from alumina phases of low crystallinity, such as aluminum hydroxides, aluminum oxide hydroxides, or transition alumina. These two paths can be discriminated by the crystal structure of the respective oxide hydroxide (cf. Figure 1). Diaspore, commonly designated α-AlO(OH), crystallizes in a hexagonal structure, while γ-AlO(OH), known as boehmite, exhibits a crystal structure with O^2−^-ions in cubic packing [1]. Although this difference in crystal structure of the oxide hydroxide modifications might not seem very important, it causes a significant and major divergence in the course of phase transformations with rising temperature. While (hexagonal) diaspore transforms directly into (hexagonal) α-Al_2_O_3_ at relatively low temperatures of about 450–600 °C, all other phase transformation cascades pass from various aluminum hydroxides through boehmite and various (cubic) transition alumina. Their respective enthalpies of formation (Δ*H_f_*) and transition to α-Al_2_O_3_ (Δ*H_→α_*) are listed in Table 1, illustrating their transient nature [2,3]. In a penultimate step, θ-Al_2_O_3_ is usually attained, which then transforms into thermodynamically stable α-Al_2_O_3_ at about 1000–1200 °C [9]. This θ→α-transition marks the final drop in enthalpy to attain the thermodynamic pit in the Al_2_O_3_ system [2,3]. It also usually brings about a significant loss in porosity and, consequently, specific surface area. Chang et al. published a review on critical (θ-Al_2_O_3_) and primary (α-Al_2_O_3_) crystallite sizes during the θ→α-transition in 2001 [10]. Their own experiments showed a critical crystallite size of 22 nm for θ-Al_2_O_3_ in order to undergo phase transformation to α-Al_2_O_3_ with resulting primary crystallite sizes of 17 nm. This dependence on the transition alumina crystallite size originates in the pseudomorphic character of the boehmite→γ-transition: One boehmite crystallite subdivides into several γ-Al_2_O_3_ crystallites, yet the eventual α-Al_2_O_3_ crystallite resumes the dimensions of the original boehmite grain, as illustrated in Figure 2 [11]. Schaper et al. provided another study of the corresponding γ→α-transition mechanism in 1985, postulating that the loss of specific surface area is not necessarily a result of but usually precedes the formation of α-Al_2_O_3_ at elevated temperatures [12]. They draw their conclusion upon the “entirely different” values obtained for activation energies of sintering (*E_A_* = 250 kJ/mol, corresponding to surface diffusion, i.e., loss of specific surface area) and nucleation (*E_A_* = 600 kJ/mol, corresponding to formation of α-Al_2_O_3_), respectively. This theorem is strongly backed by the findings of numerous groups on low temperature formation of corundum from diaspore, i.e., without sintering effects and the like, as shall be explicated in the following section.

In fact, a superior stability of transition alumina for smaller particle sizes, i.e., higher specific surface areas, was already suggested by McPherson in 1973 [13]. In a more general manner, McHale, Perrotta, and Navrotsky illuminated the thermodynamic stability of nanocrystalline γ- vs. α-Al_2_O_3_ with respect to the specific surface area in 1997 [14,15]. Their findings and calculations suggest that for *A_BET_* > 175 m^2^/g, γ-Al_2_O_3_ might in fact be the thermodynamically stable modification of Al_2_O_3_. This would in turn imply an impossibility for α-Al_2_O_3_ to exhibit specific surface areas larger than 175 m^2^/g above a certain threshold temperature, which is not explicated but likely below the formation temperature of either Al_2_O_3_ modification. In an extension of this work published in 2015, Navrotsky even postulates that this limiting specific surface area amounts to only 130 m^2^/g for α-Al_2_O_3_, and that for *A_BET_* > 370 m^2^/g, i.e., very small particle sizes, amorphous alumina is more stable than either crystalline modification [16]. All but one of the articles cited in the following section seem to prove her and her co-workers right.

### 2.2. Diaspore-Derived Corundum

The literature treating diaspore syntheses dates back to 1943, when Laubengayer and Weisz first reported on the successful preparation of diaspore in the laboratory [17]. They synthesized diaspore under various hydrothermal conditions, starting from boehmite, γ-Al_2_O_3_, or corundum. However, they were only able to obtain diaspore when seeding the starting material with naturally occurring diaspore crystals. In 1960, Torkar then claimed the first synthesis of diaspore without seeding crystals [18]. Wefers prepared diaspore hydrothermally at only 100 °C by coprecipitating iron and aluminum hydroxide gels, reducing the nucleation energy for diaspore [19]. However, synthesizing diaspore with or without seeding crystals remains a challenging and oftentimes protracted undertaking, as evidenced by the small number of publications thereunto and the elaborate experimental descriptions therein [20,21].

Having successfully synthesized diaspore, one may transform it into α-Al_2_O_3_ by thermal decomposition at about 500 °C, i.e., relatively mild conditions [21,22,23,24,25,26,27]. Specific surface areas as high as 150 m^2^/g [26] or even 160 m^2^/g [28] can be obtained in the resultant high specific surface area (HSSA) α-Al_2_O_3_. Mitchell even claims specific surface areas of up to 600 m^2^/g in his 1977 patent, though without giving a concrete example thereto [29]. However, when thermal decomposition of diaspore into corundum is executed at only slightly higher temperatures (> 600 °C), *A_BET_* decreases to values below 10 m^2^/g [22,26,28]. In fact, Wefers contested complete transformation of diaspore into corundum at lower temperatures, postulating a certain fraction of less well-ordered Al_2_O_3_ in diaspore-derived corundum up to temperatures of 1000 °C [30]. So, despite its thermodynamically stable crystal structure, its porosity is not inert towards sintering effects at relatively low temperatures. This renders diaspore-derived low temperature HSSA unsuitable for most catalytic applications, since the difficulties occurring in practice are similar to those one encounters with transition alumina, e.g., loss of specific surface area and porosity, sintering and consecutive deactivation, or loss of washcoat material due to thermal stress [5,31].

### 2.3. Hydrothermal Syntheses

Hydrothermal methods represent a straightforward and relatively green way to synthesize inorganic materials. Perrotta provided a review on the efforts made on nanosized corundum synthesis, including hydrothermal methods [28] in 1998. In 2010, Suchanek published a number of articles on hydrothermal syntheses of α-Al_2_O_3_ powders [32], nanosheets [33,34], and nanoneedles [33]. The conditions were 430–450 °C at 10.3 MPa in order to circumvent diaspore formation [32]. Boehmite powders were converted into pure α-Al_2_O_3_ assisted by commercial α-Al_2_O_3_ seeds. While the *A_BET_* of the produced powders diminished to 7 m^2^/g upon heating to 1000 °C, while nanosheets, whose morphology was modified by addition of nanosized colloidal silica, preserved their specific surface area of 20 m^2^/g even upon heating.

Ghanizadeh et al. then followed with a comparative study on hydrothermal and precipitation routes [35]. Unsurprisingly, their seeded precipitates converted into α-Al_2_O_3_ at lower temperatures, while they managed to convert seeded boehmite powder into corundum under hydrothermal conditions at 220 °C for 24 h. However, this does not hold true for unseeded boehmite; specific surface areas for these samples are not included. Precipitated powder calcined to pure α-Al_2_O_3_ at 1200 °C for 1 h exhibits a specific surface area of 8.1 m^2^/g, with intraparticular voids around 200 nm. Similar results had already been presented by Sharma et al., claiming specific surface areas of 245 m^2^/g for seeded α-Al_2_O_3_ powder hydrothermally synthesized at 190 °C [36]. Their XRD data clearly show an incomplete conversion into corundum, which explains the unrealistically large specific surface area.

Even more recently, Yamamura et al. reported a hydrothermal method combined with a seeding technique, showing α-Al_2_O_3_ reflexes in XRD at temperatures as low as 400 °C, yet without reaching complete conversion to α-Al_2_O_3_ even for 5 wt.% seeding crystals [37]. A second publication shortly after shows XRD reflexes for α-Al_2_O_3_ starting from 600 °C [38]. However, complete conversion to α-Al_2_O_3_ is still not achieved until annealing temperatures reach at least 900 °C. The obtained particle sizes are in the sub-micrometer range, whereof relatively large specific surface areas may be inferred, although numbers are not presented.

Different approaches using aluminum precursors with carbonaceous anions also may yield porous α-Al_2_O_3_. Lee at al. obtained α-Al_2_O_3_ with an *A_BET_* of 25 m^2^/g at 900 °C with hydrothermally synthesized aluminum oxalate hydroxide precursors [39], while Ahmad et al. recently reported on α-Al_2_O_3_ from hydrothermally prepared ammonium aluminum carbonate hydroxide whiskers with distinct pore diameters of 260 nm, yet without precisely determining the corresponding pore volume [40,41].

### 2.4. Pore Protection by Carbon Filling

A different approach towards porous α-Al_2_O_3_ is found in the attempt to prevent pore collapse upon high temperature calcination by infiltrating porous γ-Al_2_O_3_ with a carbonaceous precursor solution. Carbon filling of porous materials is usually employed in syntheses of porous carbon materials and can easily be realized with different carbonaceous precursors [42,43,44]. Utilizing this method not to template a porous carbon but to protect the pores of a ceramic material such as Al_2_O_3_ has been patented by Shell in 1971 [45] and seconded by Exxon in 1978 [46]. Both inventors claim specific surface areas in α-Al_2_O_3_ of 20–70 m^2^/g, though without providing XRD data to prove complete transformation of the previously infiltrated γ-Al_2_O_3_. An observed decrease in specific surface area upon heat treatment at 1100 °C is argued to be due to the “surface area stability of the alpha alumina” lying between about 1000 °C and 1100 °C. We consider an incomplete θ→α-transition much more likely to be the cause thereof, as is also observed by the patent holders themselves on other samples [46]. In a similar process, Holler (also from Shell) claimed to attain specific surface areas of at least 40 m^2^/g by thermally decomposing a porous polymeric carboxylate chain-bridged by aluminum ions [47]. Wen et al. patented a route using oleic acid as carbon precursor to manufacture ultrafine α-Al_2_O_3_ powder with particle sizes *d* < 60 nm [48]. A biomimetic approach towards HSSA α-Al_2_O_3_ was chosen by Benítez Guerrero et al., using lignocellulosic materials both as templates and carbon source, claiming specific surface areas of 100 m^2^/g [49]. This article also contains a recommendable review on HSSA α-Al_2_O_3_ (written in Spanish). Niu et al. introduced carbon nanotubes as templates into their alumina sol [50]. Consequently, they also followed a similar procedure as in described in the above-mentioned patents, i.e., the alleged transformation from γ-Al_2_O_3_ into α-Al_2_O_3_ was performed under N_2_ flow at 1150–1300 °C, followed by pyrolytic carbon removal by oxidation at 500–750 °C. Products obtained via this method are certainly porous, and likely alumina, despite the possibility of aluminum nitride formation above 900 °C [51,52,53,54]. However, the authors of this review contest the formation of 100% pure corundum under the conditions described. Inconclusive XRD data from other groups using sucrose as carbon precursor support our reasonable doubts [55,56], as do constraints formulated by patent holders [46], results obtained by Schüth et al. [57], and in our own laboratories (cf. Section 5.5). These doubts are reasonable, as the free surface energy *γ_s_* of γ-Al_2_O_3_ amounts to only 1.5–1.7 J/m^2^, compared to 2.64 J/m^2^ for α-Al_2_O_3_ [16]. This difference in free surface energy impairs the conversion of transition alumina to corundum at specific surface areas > 175 m^2^/g [14,15], even at temperatures above 1200 °C, if the microstructure and hence the specific surface area are to be conserved. Also, kinetic hindrance [57] due to the generally much smaller crystallite grain size in γ-Al_2_O_3_ compared to α-Al_2_O_3_ [10,13] suggests that even calcination in vacuo at temperatures as high as 1300 °C does not necessarily result in conversion of carbon-filled porous transition alumina to corundum.

### 2.5. Introducing Porosity via Soft Templating in Sol-Gel Syntheses

A well-established and versatile approach to introduce tailor-made pore structures into metal oxides is found in sol-gel syntheses [58,59,60]. For alumina, alkoxides have been employed as precursors for almost a century [61,62]. The use of aluminum salts with an epoxide-assisted route was first presented by Baumann, Gash and co-workers 15 years ago [63] and has since developed into a well-established method [64,65,66,67,68,69]. Ultimately, porosity of the synthesized gels much depends on the chosen drying conditions. Xerogel granules can be obtained by evaporative drying methods, cryogels by freeze-drying [70,71], or aerogel monoliths by drying under supercritical conditions, respectively [65,72]. Whichever drying method is chosen, it cannot effectively prevent the collapse of mesopores originating from the sol-gel process upon high temperature calcination, which is necessary to accomplish the phase transition from θ- to α-Al_2_O_3_. (Phase transformation in the routes discussed here follows the boehmite path, including cubic transition alumina forms, as shown in Figure 3 [68] (cf. also Figure 1, adapted from [9].).

Polymer induced phase separation is a widely used strategy in sol-gel syntheses [73,74,75]. Generally speaking, it enables introduction of macroporosity into otherwise mesoporous or even non-porous materials, and for a relatively precise adjustment of macropore size. Common phase separating agents in alumina synthesis are polymers such as polyethylene oxide [64], polyacrylamide [76], or Pluronic P123 [77,78]. With the aid of P123 and additional tetrabutyl ammonium hydroxide, López Pérez et al. were able to obtain pure α-Al_2_O_3_ with *A_BET_* in the range of 16–24 m^2^/g by evaporation induced self-assembly [79], while Zhang et al. report a value of 18 m^2^/g with additional PEG8000 after calcination at 1200 °C [80,81]. Other approaches include EDTA, leading to α-Al_2_O_3_ powder with up to 14 m^2^/g [82], a combination of chelating agents with acrylamide [83], or other surfactants, for instance [84,85]. We recently reported on the use of citric acid as a phase separating agent [68], which has been known to act as a template in alumina sol-gel syntheses [86,87,88,89,90] but never brought to comparable porosity values in α-Al_2_O_3_ before. Not only does it enable pore volumes of up to 1.2 cm^3^/g with macropores ranging from 0.1–5.3 µm but *A_BET_* were also increased up to 12 m^2^/g with the aid of citric acid. Similar effects can be achieved by addition of dicarboxylic acids, though with even better tunability of pore widths [91].

### 2.6. Porous Membrane Preparation via Anodic Oxidation of Aluminum

Self-ordered anodized aluminum oxide (AAO) is a highly ordered porous alumina with hexagonal tubular pores. It can be produced via an anodization process under specific reaction conditions, yielding pore sizes in the range of 10–400 nm. Masuda et al. first presented a two-step anodization process to obtain these ordered membranes in 1995 [92,93]. Hillebrand et al. presented a methodology for analyzing and quantifying the grain morphology of self-ordered porous AAO membranes based on interpore distance and angular order [94]. Other publications treat the fabrication of tubular membranes [95], thin films (50 µm thickness) [96], or three-step anodization [97]. Lee and Park provided a comprehensive review on the matter of AAO in 2014 [98]. These as-synthesized membranes are amorphous and hence highly labile to both acid and base attacks. Heat induced crystallization of porous AAO leads to changes of mechanical properties but also in morphology of the porous system [99,100,101]. There is no acknowledged routine thermal treatment for the conversion of amorphous AAO membranes into α-Al_2_O_3_ without serious mechanical deformation [99,100,102]. Chang et al. prevented severe deformation of the porous system by removing the acid-anion contaminated outer pore wall oxide before annealing at high temperatures [103]. This selective removal of the outer pore walls tremendously increases the pore diameter, while simultaneously reducing the walls’ thickness. A successful approach of transforming amorphous membranes into porous α-Al_2_O_3_ was published in 2018 by Hashimoto et al. [104,105]. They preserved the initial pore structure of *d_p_* ≈ 220 nm, introduced by anodization with phosphoric acid, after calcination at 1400 °C, and, moreover, were also able to subsequently remove AlPO_4_ nanoparticles from the pore walls, leaving behind a second pore system in the mesopore range (*d_p_* ≈ 20 nm). This increased the specific surface area from 3.5 m^2^/g in the as-synthesized membranes to 11.7 m^2^/g in the heat- and HCl-treated ones.

### 2.7. Ultrafine α-Al_2_O_3_ Powders with High Specific Surface Area

Instead of aiming for porous α-Al_2_O_3_, one may also attempt to render particle sizes as small as possible in order to maximize the specific surface area of a material, which in this case is not exactly congruent with the internal surface area, as is often suggested in literature. We provide an algebraic estimate of attainable specific surface areas via this approach in Section 4.2.

For α-Al_2_O_3_, there have been a number of different methods suggested to this end in literature. In 1963, the Allied Chemical Corporation of New York patented an elaborate process involving direct oxidation of aluminum chloride vapors. The resulting α-Al_2_O_3_ particles are claimed to not exceed 5000 nm in size, yielding a specific surface area of 6.2 m^2^/g [106]. Degussa AG patented a process starting from γ-Al_2_O_3_ powder, yielding α-Al_2_O_3_ particles of 100–200 nm only, and specific surface areas of some 40 m^2^/g. However, the α-Al_2_O_3_ content in the final product is claimed to amount to only 70%–90% [107]. A third patent by the Taiwanese National Science Council utilizes oleic acid as an additive to obtain α-Al_2_O_3_ particles of *d* < 60 nm; *A_BET_* values are not provided [48].

By combustion of a pasty mixture of Al(NO_3_)_3_ 9 H_2_O and urea at 500 °C, Kingsley and Patil obtained a foam-like mass which spontaneously “ruptures with flame and glows to incandescence”, with temperatures of 1600 °C [108]. The resultant product is a voluminous and foamy α-Al_2_O_3_ with an *A_BET_* of 8.3 m^2^/g, wherefrom they calculate a particle size of 220 nm. Ogihara et al. presented their precipitation route starting from an aluminum alkoxide in different solvents in 1991, yielding particle sizes down to 250 nm [109]. Zeng et al. prepared α-Al_2_O_3_ of only 25–30 nm by precipitating boehmite from an aqueous solution of AlCl_3_ with NH_3_ and acetic acid as peptizing agent, followed by freeze-drying of the resulting powder [110]. They measured a specific surface area of 51 m^2^/g for their essentially pure α-Al_2_O_3_ powder.

Since the turn of the century, a number of different studies have been published, including emulsion evaporation [27,111,112], precipitation methods [35,113,114,115], solution combustion [116], even attaining a value of 54 m^2^/g for *A_BET_* [117] (but incomplete α-transition), also using a microwave oven [118], microwave-assisted [119] or α-seeded sol-gel synthesis [120], or direct combustion of aluminum hydroxide acetate [121], for instance. Sharma et al. studied the role of pH in alumina sol-gel synthesis [122]. They were able to obtain small particles (55–70 nm) at pH 2.5 and identify α-Al_2_O_3_ by XRD at 930 °C already. Since phase transformation appears to be incomplete, the reported specific surface area of 130 m^2^/g seems to be unlikely, though. Zhang et al. were able to synthesize α-nanorods with diameters as small as 5 nm [123,124]. However, despite this extremely favorable morphology, they only obtained a specific surface area of 8.2 m^2^/g. Zaki et al. also reported cylindrical particles of 100–200 nm in length and less than 25 nm in diameter, yielding 18 m^2^/g for *A_BET_* [125]. They used citric acid and acrylic acid in a modified Pechini process, yet obtaining not entirely pure α-Al_2_O_3_ particles at 900 °C. Recently, Petrakli et al. published a route for the obtention of α-Al_2_O_3_ nanospheres < 10 nm from aqueous suspensions of nano-boehmite, stabilized by hyperbranched dendritic poly(ethylene)imine [126]. By calcination at 1050 °C, they attained a specific surface area of 14 m^2^/g in their pure α-Al_2_O_3_ powder.

### 2.8. Miscellaneous Methods

Standard synthesis options of α-Al_2_O_3_ naturally include direct conversion of boehmite without any special precautions. There are again sol-gel syntheses, one example yielding an *A_BET_* of 9 m^2^/g [127], or supposedly even 12 m^2^/g after calcination in nitrogen [128], although both cited results do not provide definitive proof of a complete α-transition and should be challenged.

Drying gels in a supercritical fluid such as methanol or ethanol may result in pure α-Al_2_O_3_ aerogel monoliths with specific surface areas of 10 m^2^/g [129]. Kinstle and Heasley even claim 30–100 m^2^/g in their 1989 patent, yet the provided XRD data suggest an incomplete α-transition [130].

Less often encountered are molten salt syntheses to produce platelets of about 2 µm × 0.1 µm by thermal decomposition of a precursor salt such as polyaluminum chloride, ammonium aluminum carbonate hydroxide, or NaAlO_2_ [131,132,133,134]. However, BET surface areas attainable for these dimensions would still be relatively low.

An extremely high and overall plausible *A_BET_* of 41 m^2^/g was recently reported by Bhattacharyya et al. [135,136]. By acid leaching of Indian kaolin, a mineral containing 30%–40% Al_2_O_3_, they obtained a solution rich in aluminum. Further concentration and addition of PVA lead to the formation of alumina beads of 2–3 mm by within oil droplets. These beads were then calcined at 1100 °C to yield XRD-pure α-Al_2_O_3_, supposedly containing mesopores of *d_p_* = 8 nm. While all data on phase composition and porosity seem to confirm this extraordinary result, the provided SEM images show much larger pores in the α-Al_2_O_3_ beads.

Another rarely mentioned method represents the mechanochemical transformation of alumina powders into α-Al_2_O_3_ [137,138,139]. Mechanochemical approaches are generally underrepresented with respect to photochemistry and thermochemistry, which is most often the method of choice. Yet by physically grinding powders, or mixtures thereof, phase transitions or chemical reactions may be induced under relatively mild conditions [140]. Applying this concept to transition alumina powders permits the obtention of α-Al_2_O_3_ by high energy milling, conveying the necessary energy for phase transitions purely in a mechanical manner. Progression of phase transitions as well as specific surface areas then become functions of milling speed and duration, as Dynys and Halloran reported, for instance [141]. It goes without saying that this is not a proper means to introduce or preserve porosity, however.

For the sake of completeness, just a brief glance shall be taken on molding procedures enabling the fabrication of monolithic high specific surface area α-Al_2_O_3_ carrier materials. This matter would in fact require a separate review article. Sintered extrudates by Wang et al. still exhibit porosities of some 50%, although no numeric evaluation of the pores is given [142]. Direct foaming of α-Al_2_O_3_ powder with polyurethane precursors led to monoliths with porosities < 90% and preservation of pre-existent pores in the alumina grains in the range of 300–500 nm in our group [143]. Nettleship and Sampathkumar produced porous compacts with specific surface areas of 20–80 m^2^/g from diaspore-derived α-Al_2_O_3_ powders by Alcoa [144], similar to the process patented later by the Japanese Yazaki Corporation [145]. Vijayan et al. used the same precursor to fabricate alumina foams by thermo-foaming of powder dispersions in molten sucrose [146,147]. Recently, we first reported the successful transformation of porous sol-gel α-Al_2_O_3_ into reticulated ceramic foams with strut porosities in the sub-micrometer range by a replica route [148].

### 2.9. Dopants Favoring or Inhibiting α-Transition

Although this article focusses on pure α-Al_2_O_3_, we shall not leave the potential of dopant metals unmentioned. Mainly due to lattice distortion effects, a number of different metals may attain quite diverse effects regarding the θ→α-transition.

Lanthanum and cerium are probably the most well-known dopants that are even applied industrially to retain large γ-Al_2_O_3_ specific surface areas at elevated temperatures [12,149,150,151,152,153,154,155,156,157,158,159]. The doping atoms are incorporated into the alumina structure and generate structures with aluminum ions in tetrahedral symmetry, hence stabilizing the γ-Al_2_O_3_ spinel structure and shifting its phase transition to higher temperatures [155,160]. Similar results can be obtained by adding calcium, thorium [160], silicon [161] or zirconium [160,162,163,164], although these two will not readily integrate into the alumina structure but rather form a distinct second phase. Nonetheless this secondary phase postpones α-transition to higher temperatures, retaining larger specific surface areas, as well.

On the other hand, a contrary effect is observed for manganese [165,166,167,168,169], iron [170,171,172], indium, and gallium [160]. Tsyrulnikov’s group suggests a solid solution of Mn^3+^ in α-Al_2_O_3_, significantly lowering the temperature required for a complete θ→α-transition [169]. This mechanism has been studied in detail and transferred to the preparation of highly ordered porous anodized α-Al_2_O_3_ membranes [168], and sol-gel alumina (cf. Section 3.1.1, Section 3.1.4 and Section 5.4) in our group.

### 2.10. Pure α-Alumina with Pores in the Sub-Micrometer Range

Only a few publications were identified showing pure α-Al_2_O_3_ with properly characterized macropore systems, some of which were already mentioned. Sokolov et al. were the first to provide a remarkable route for macroporous α-Al_2_O_3_ powder in 2003 via a sol-gel route and templating with poly(methyl-methacrylate) colloidal crystals [173]. They attained pore diameters from 300–1000 nm and *A_BET_* values from 10–24 m^2^/g. Ten years later, Wang et al. investigated photoluminescence in their porous monolithic α-Al_2_O_3_ prepared by a modified polyacrylamide gel route, finding pore diameters of 80–500 nm [83]. Topuz et al. reported pore diameters of 110 nm in macroporous α -alumina supports prepared by using vacuum assisted filtration of α-Al_2_O_3_ suspensions [174]. Finally, three articles were published in 2019: Ahmad et al. reported on α-Al_2_O_3_ from hydrothermally prepared ammonium aluminum carbonate hydroxide whiskers with distinct pore diameters of 260 nm, yet without precisely determining the corresponding pore volume [41]. Quite an interesting morphology in sol-gel α-Al_2_O_3_ was reported very recently by Roque-Ruiz et al. They obtained fibers with diameters of 230–900 nm by electrospinning an aluminum nitrate precursor solution [175]. After sintering at 1600 °C, the alumina presented macropores in the sub-micrometer range, though with small pore volumes and negligible *A_BET_*. Two of this paper’s authors earlier shared their results on macroporous α-Al_2_O_3_ via a citric acid-assisted sol-gel synthesis, yielding pore diameters in the range of 115 nm to 6 μm, with corresponding pore volumes between 0.17 and 1.19 cm^3^/g. *A_BET_* values attained up to 12 m^2^/g [68].

## 3. Materials and Methods 

### 3.1. Syntheses of Macroporous α-Al_2_O_3_

Several methods were employed to introduce macroporosity into high temperature stable alumina. Sol-gel routes and anodic oxidation start from precursors containing Al, generating the aluminum oxide in situ (Section 3.1.1, Section 3.1.2 and Section 3.1.3). In other approaches, already available γ-Al_2_O_3_ samples (from industrial provenance or synthesized in our group) were treated by either carbon infiltration (Section 3.1.5), or impregnation with precursor solutions containing Mn or Fe, and subsequent calcination (Section 3.1.4).

Unless stated otherwise, reagents were used as received, without further purification or treatment. Ethanol and distilled water were taken from domestic lines. AlCl_3_·6H_2_O (99% purity) was purchased from Alfa Aesar (Haverhill, MA, USA), citric acid (food quality) from Purux (Schwarzmann GmbH, Laaber, Germany), and all employed dicarboxylic acids from Merck (Darmstadt, Germany). Aluminum tri-*iso*-propoxide, propylene oxide (99%) and polyethylene oxide (PEO, *M_W_* 900,000) were delivered by Acros Organics (Geel, Belgium). Ultra-pure (>99.999%) aluminum chips were provided by EVOCHEM Advanced Materials (Offenbach am Main, Germany).

#### 3.1.1. Epoxide-Mediated Sol-Gel Synthesis of Porous α-Al_2_O_3__re_

For a standard procedure synthesis, 7.80 g of AlCl_3_·6H_2_O and 6.98 g of distilled water were placed in the reaction vessel and dissolved in 7.90 g of ethanol, resulting in an r-value (ratio of water molecules to Al-ions) of 18. Different additives were used as porogenes: PEO (*M_w_* 900,000) was dissolved with the educts, while di- and tricarboxylic acids were added immediately before the reaction. The respective amounts are listed inor can be drawn from [68] and [91]. The solution was placed in an ice bath and cooled down to 4 °C while stirring. Propylene oxide (7 mL) was then added with a syringe under vigorous stirring. The ice bath was removed after three minutes. After a total reaction time of 10 minutes, the stirring bar was taken out of the reaction vessel, which was then placed in a 40 °C water bath for 24 h to allow for gelation and gel ageing. Subsequently, solvent exchange took place in an ethanol bath for 3 days succeeded by an acetone bath for another 3 days. After drying at 70 °C for 7 days, calcination followed at 1200 °C (heating rate 3 K/min) for 6 h to obtain pure porous α-Al_2_O_3_.

Samples prepared via the epoxide-mediated sol-gel process are designated SG-, followed by a specification of the additives used, and in two cases a modified ageing temperature of 80 °C.

#### 3.1.2. Mutual Cross-Hydrolysis in Combined Sol-Gel Synthesis

This synthesis route is inspired by a bimetallic sol-gel synthesis of yttria–alumina [176], which uses an aluminum alkoxide and yttrium salts. We transferred this approach to the synthesis of pure alumina, also employing an aluminum salt along with the alkoxide. As both reagents simultaneously hydrolyze each other, the use of an epoxide or base becomes unnecessary.

Two different approaches were tested to this end. On the one hand, samples with added water, designated MCH-w, were prepared by mixing the standard reaction solution employed for the epoxide-mediated sol-gel process with an aluminum alkoxide. For a standard MCH-w sample, 3.43 g AlCl_3_·6H_2_O were dissolved in 3.07 g distilled water and 3.48 g ethanol. A mass of 2.90 g of aluminum tri-*iso*-propoxide Al(OC_3_H_7_)_3_ (AIP), corresponding to 1.0 molar equivalents, were added and stirred vigorously until complete dissolution. The stirring bar was removed, and the closed vessel was then placed in an oven set to 80 °C. Gelation occurred under static conditions within a few hours. After opening the vessels, samples were left in the oven for ageing for 3 days.

In contrast, samples prepared only in organic solvents, designated MCH-o, were synthesized by first preparing a solution of 3.00 g Al(NO_3_)_3_·9H_2_O in 7.5 g ethanol, and a solution of 2.97 g of aluminum tri-*sec*-butoxide Al(OC_4_H_9_)_3_ (ASB) in 7.5 g *iso*-propanol, corresponding to a molar ratio of 3:2 for ASB/Al(NO_3_)_3_. Both solutions were combined under vigorous stirring for three minutes, then placed in an oven at 80 °C for gelation. Samples were dried for 1 day in the same oven.

All obtained gels were then calcined at 1200 °C for 6 h, yielding pure porous α-Al_2_O_3_.

#### 3.1.3. Anodic Oxidation of Aluminum Chips

The typical process for the formation of self-ordered anodic aluminum specimen via two-step mild anodization of ultra-pure (>99.999%) aluminum chips has been described in literature [92,177,178]. Ultra-pure aluminum chips provided by EVOCHEM Advanced Materials, with a thickness of 0.5 mm and a diameter of 20.0 mm, were pretreated by several washings with ultra-pure solvents like ethanol, iso-propanol, acetone and deionized water. The cleaned chips were annealed in a furnace for 3 h at 500 °C. Before being mounted onto the anodization setup, with the aluminum chip serving as anode and a platin wire as cathode. The surfaces of the chips were electro-polished at room temperature under moderate stirring for 20 minutes at 25 V in a mixture of a 60% HClO_4_ aqueous solution and ethanol. To start the anodization process, the setup was rinsed with deionized water and refilled with the electrolyte solution (0.3 M oxalic acid). Afterwards the setup was cooled down and voltage was applied (40 V). After 24 h anodization and a constant current flow (15–25 mA) the voltage was stopped, the anodized surface was washed with deionized water and treated with a solution of CrO_3_ in aqueous H_3_PO_4_ to selectively remove the alumina from the aluminum chip. A second anodization step was started with fresh electrolyte solution under the same conditions as applied previously. Depending on the anodization duration, various thicknesses of alumina can be obtained [92]. Remaining aluminum was removed by an etching process with a solution of CuCl_2_ and concentrated aqueous HCl. After completion of the reaction, opaque alumina membranes without metal residues were obtained, which were washed in deionized water and dried before further treatment [168].

#### 3.1.4. Solid Solutions Concept

Manganese-assisted α-transition of alumina samples was inspired by Tsyrulnikov’s work [169]. The manganese precursor was deposited on AAO membranes or sol-gel samples by immersing in 1 M Mn(NO_3_)_2_ solution for 5 minutes. Excess solution was removed, and the membrane was pre-annealed at 200 °C at 1 K/min to combust the nitrates and obtain a homogenous surface coverage of manganese oxides. Calcination at higher temperatures (≥900 °C) led to phase transition, with an earlier onset of α-alumina formation. The same procedure was executed using a 1 M solution of Fe(NO_3_)_3_.

Manganiferous species formed during this procedure were extracted afterwards by acid leaching. Calcined samples were placed in concentrated aqueous HCl and stirred for four hours, until their color had changed from brown to a light rose.

For comparison, sol-gel syntheses were also executed but with 5 mol-% of MnCl_2_·4H_2_O (and adjustment of water content), or FeCl_3_·6H_2_O, respectively. The remaining 95 mol-% were accounted for by AlCl_3_·6H_2_O, as described above in Section 3.1.1. The degree of phase transition was then evaluated after calcination at 950 and 1050 °C.

#### 3.1.5. Pore Protection by Carbon Filling

Two different kinds of θ-Al_2_O_3_ samples served as starting materials for this route: extruded pellets by Alfa Aesar, and sol-gel granules (2–5 mm) synthesized in our lab, as described in Section 3.1.1. They were converted into pure θ-Al_2_O_3_ by calcination at 1000 °C for 6 h, and were treated as follows: Firstly, sucrose was dissolved in water to yield a solution of 68 wt.%. This concentration is just below the point of saturation, in order to ensure proper flowing behavior of the solution. Alumina samples were then placed in a flask and covered with sucrose solution. During evacuation of the flask, pores filled with sucrose solution. Drying at 120 °C for 1 h was followed by carbonization at 650 °C for 1 h under constant nitrogen flow. This step was repeated until mass constancy. Samples with carbon-filled pores were then calcined at temperatures between 1150 and 1350 °C under vacuum (12 mbar). These temperatures are sufficient to enable phase transition to α-Al_2_O_3_ under standard conditions. Remaining carbon was removed by pyrolysis in an oven at 700 °C.

#### 3.1.6. List of Samples

Table 2 below gives an overview of all samples from our own labortatory discussed in the calculation and results sections, with references to the experimental and discussion sections.

### 3.2. Characterization Techniques

Structural and textural characterization of the samples discussed below was conducted by mercury intrusion, scanning electron microscopy (SEM) coupled with energy-dispersive X-ray spectroscopy (EDX), and nitrogen sorption, as well as X-ray diffraction (XRD). For manganese-impregnated samples, inductively coupled plasma optical emission spectroscopy (ICP-OES) was also used to determine the Mn fraction.

Mercury intrusion was performed with a Pascal 440 porosimeter by ThermoScientific/Porotec with pressures ranging from 0.2 mbar to 4000 bar. Mercury surface tension was assumed to be 0.484 N/m, its contact angle was set to 141.3°. Samples were outgassed at 0.2 mbar for 10 minutes at room temperature prior to filling the dilatometer with mercury.

SEM images were obtained using two different devices, a Leo Gemini 1530 by Zeiss (Oberkochen, Germany) with an Everhart-Thornley detector, and a Nova Nanolab 200 with a TLD Elstar detector (both by FEI, Hillsboro, OR, USA) for collecting secondary electrons, respectively. EDX measurements were also carried out on the latter device, using a SUTW-Sapphire detector (EDAX Inc., Mahwah, NJ, USA). Samples were fixated on a carbon foil and vapor coated with a gold film. Accelerating voltage was 10 kV for both devices.

SEM images of AAO membranes were also subject to image analysis with ImageJ (Version 1.52, retrieved 24/05/19, National Institute of Health, Bethesda, MD, USA) for pore detection and MATLAB (Version R2017a, retrieved 03/05/19, T.M.Inc., Natick, MA, USA) for calculation and data analysis. An in-depth description of these analyses is omitted here for reasons of conciseness but will be given in [168].

Nitrogen sorption (ASAP 2000, Micromeritics, Norcross, GA, USA) was used exclusively to determine the specific surface area (*A_BET_*). Prior to examination, the samples were dried and activated at 300 °C under ultrahigh vacuum. The determination of specific surface areas was conducted using the linearized form of the BET equation in the range of 0.05 ≤ p/p_0_ ≤ 0.30 [179].

The degree of transformation to α-Al_2_O_3_ and phase composition were confirmed by XRD, either on a D8 Discover by Bruker (Billerica, MA, USA) with a VANTEC500-2D GADDS detector, using CuK_α_ at 40 kV and 40 mA, or on a SmartLab by Rigaku with a HyPix-3000 2D HPAD detector, also using CuK_α_. Interpretation and phase identification were conducted using *Match!* software by Crystal Impact, Bonn Germany.

ICP-OES samples were pre-treated in a mixture of strong acids in a Multiwave 3000 microwave oven by Anton Paar at 1300 W. ICP-OES analysis was then carried out using an OPTIMA 8000 by Perkin Elmer (Waltham, MA, USA).

## 4. Calculation of Theoretical Porosity Limits

This section gives some theoretical calculations as to which values of pore diameters and volumes, or particle sizes, are required to attain a certain internal surface area, i.e., *A_BET_*. We consider two different possibilities of maximizing the specific surface area, namely rendering a bulk material as porous as possible, or minimizing the particle size of a (non-porous) powder sample.

These estimative calculations are then related to values that have actually been reported. We also provide a projection of what *A_BET_* values can realistically be expected for α-Al_2_O_3_.

### 4.1. Porous Material

The following assumptions shall be made:the pore diameter corresponds to the modal pore diameter, *d_p,mod_*,the pore diameter is constant over the entire length of a pore, meaning thatthe pore is a perfect cylinder with
(1)$$V=\pi hr^{2},\;\mathrm{or}\;h=\frac{V}{\pi r^{2}}$$
where *V* is the volume, *h* is the height, and *r* is the radius of the cylinder,all pores of a mode have the same diameter.

We can now calculate a theoretical internal surface area *A_theo_*, taking the sum of all pore lengths as the height of a single cylinder with *d_p,mod_*, and using the pore volume *V_p_* (cf. Equation (2)). Since the pores are open towards the surface, we do not need to consider the areas of the circular lids. (*NB:* In any case, these areas would be negligible next to the length of the pores.)
(2)$$A=2\pi hr=2\pi \frac{V}{\pi r^{2}}r=\frac{4V_{p}}{d_{p,mod}}$$

The density *ρ* of the material in question is already included in this calculation via the pore volume, given in [cm^3^/g] and hence normalized to the sample mass. (For α-Al_2_O_3_, *ρ* = 3.997 g/cm^3^ [180]).

Table 3 gives an overview of calculated internal surface areas for several α-alumina samples from Section 5.1. Experimental *A_BET_* are juxtaposed and their ratio is calculated, giving an account of how close the respective sample is to the calculation of the ideal system. In evaluating the results, we need to consider that even small errors have a large effect on BET calculations from nitrogen sorption measurements at such low values. In view of the limit of the method, which is commonly supposed to be around values of 10 m^2^/g, the accordance of calculated and BET surface areas is in an acceptable range. This is also indicated by the deviation ratios |*A_BET_* − *A_theo_*|/*A_BET_*, which remain below 0.20 for most samples. The three samples listed at the bottom of 3 do not seem to comply with the theory. The deviations for samples SG-C2#1 and SG-C6 are still within a range that can be explained by experimental errors, due to the extremely low *A_BET_* values. In sample SG-CA19, there must be a different effect, possibly the tortuosity and superficial roughness within the pores. Nonetheless, our calculations appear to be apt to predict the specific surface area of sol-gel α-alumina samples within a certain margin.

As demonstrated below in Section 5.4, the minimum pore diameter attained in α-Al_2_O_3_ is 44 nm, whereas the largest volume amounted to 1.53 cm^3^/g for sample SG-C2#2 Section 5.1. If it were possible to achieve a product combining these two optimized values, the internal surface area would amount to 139 m^2^/g. Although this value is an unsustainable overestimate, it suggests that a specific surface area of 100 m^2^/g is not entirely impossible. However, we would like to propose a more realistic estimate, considering 150 nm as a reasonably attainable pore diameter in α-Al_2_O_3_, with a pore volume of no more than 1.0 cm^3^/g at this value (cf. Section 2.10, Section 5.1 and Section 5.2). The resulting internal surface area would amount to 27 m^2^/g, a value that has almost been reached by Sokolov et al. with their reported 24 m^2^/g [173]. Any combination of smaller pore diameters with even larger pore volumes appears infeasible. Based upon an extensive literature research, years of laboratory work, and estimative calculations, the authors hence project that there will likely be no porous pure α-Al_2_O_3_ item significantly exceeding these values.

### 4.2. Nanopowder

For nanopowder, we shall consider two different models as to the assumed particle shape. Firstly, we will demonstrate how spherical particles, having the lowest possible specific surface area for a certain particle size, would affect the overall specific surface area of a powder. In a second step, particles will be considered to be tetrahedral, exhibiting the highest specific surface area of the Platonic solids [181], normalized to the same particle mass.

The following assumptions shall be made:particles are entirely non-porous,all particles exhibit the same diameter, implying a perfectly monomodal size distribution,particles are either
(3)$$a\;\mathrm{perfect}\;\mathrm{sphere}\;\mathrm{with}\;A_{s}=4\pi r^{2}=\pi d^{2}$$
(4)$$\mathrm{and}\;V_{s}=\frac{4}{3}\pi r^{3}=\frac{\pi }{6}d^{3}$$
(5)$$\mathrm{or}\;a\;\mathrm{perfect}\;\mathrm{tetrahedron}\;\mathrm{with}\;A_{t}=\sqrt{3}a^{2}$$
(6)$$\mathrm{and}\;V_{t}=\frac{a^{3}}{6\sqrt{2}}$$comparison between the two models is based on uniform particle volume, i.e., constant mass.

In order to normalize with respect to the mass of a particle, we shall now determine the particle mass *m_p_* as a function of its diameter *d* (sphere, Equation (7)), or edge length *a* (tetrahedron, Equation (9)), respectively:


**1st case: sphere**
(7)$$m_{p,s}=f\left(d\right)=V_{s}\times \rho =\frac{\pi }{6}d^{3}\times \rho$$


By dividing the surface area of a spherical particle by its mass, we obtain from Equation (7) the specific surface area of the material made up of spherical particles with diameter *d*, as given in Equation (8):(8)$$A_{s}^{spec}=\frac{A_{s}}{m_{p,s}}=\frac{\pi d^{2}}{\frac{\pi }{6}d^{3}\times \rho }=\frac{6}{d\times \rho }$$


**2nd case: tetrahedron**
(9)$$m_{p,t}=f\left(a\right)=V_{t}\times \rho =\frac{a^{3}}{6\sqrt{2}}\times \rho$$


By dividing the surface area of a tetrahedral particle by its mass, we obtain from Equation (10) the specific surface area of the material made up of tetrahedral particles with edge length *a*:(10)$$A_{t}^{spec}=\frac{A_{t}}{m_{p,t}}=\frac{\sqrt{3}a^{2}}{\frac{a^{3}}{6\sqrt{2}}\times \rho }=\frac{6\sqrt{6}}{a\times \rho }$$

Since *d* and *a* are not directly interchangeable, we opt for tetrahedral and spherical particles of the same mass (Equation (11)):(11)$$m_{p,t}=m_{p,s}$$
(12)$$\frac{a^{3}}{6\sqrt{2}}\times \rho =\frac{\pi }{6}d^{3}\times \rho$$
(13)$$a^{3}=\sqrt{2}\pi d^{3}$$
(14)$$a=\sqrt[{3}]{\sqrt{2}\pi }\times d\approx 1.644\times d$$

After inserting the known material density of α-Al_2_O_3_ from [180]
(15)$$\rho =\rho_{\alpha }=3.997\;g/cm^{3}\approx 4\times 10^{6}g/m^{3}$$
we are now able to calculate the specific surface areas both for spherical (Equation (16) follows from Equation (8)) and tetrahedral (Equation (17) follows from Equation (10)) particles of identic mass as a function of the diameter of a spherical particle:(16)$$A_{s}^{spec}=\frac{6}{d\times \rho }\approx \frac{1}{d}\times 1.5\times 10^{-6}\;m^{2}/g$$
(17)$$A_{t}^{spec}=\frac{6\sqrt{6}}{a\cdot \rho }=\frac{6\sqrt{6}}{\sqrt[{3}]{\sqrt{2}\pi }\cdot d\cdot \rho }\approx \frac{1}{d}\times 2.2\times 10^{-6}\;m^{2}/g$$

For α-Al_2_O_3_ particles with $d=1000\;nm=10^{-6}\;m$, we then obtain specific surface areas from 1.5–2.2 m^2^/g, and for particles with $d=10\;nm=10^{-8}\;m$, these values go up to 150–220 m^2^/g.

These estimates are in good agreement with experimental results from Suchanek, for example, who reports specific surface areas of 9–27 m^2^/g for grain sizes of 100–250 nm [32]. (cf. also Section 2.3) Assuming non-porous spherical particles, our calculations predict specific surface areas of 6–15 m^2^/g, or 8.8–22 m^2^/g for tetrahedral particles, respectively. Our slight underestimation compared to the literature data may well be due to a not perfectly even and non-porous surface of the particles in question.

## 5. Recent Results and Discussion

### 5.1. Epoxide-Mediated Sol-Gel Synthesis of Porous α-Al_2_O_3_

The sol-gel process is probably the most versatile approach to rendering α-Al_2_O_3_ porous. The concept of epoxide-mediated gelation of aluminiferous sols was first published in 2005 by Gash, Baumann and co-workers [63] as an adaption of their previously reported novel synthesis route for iron oxides from dissolved Fe(III) salts [72]. Tokudome et al. extended this route to polymer-induced phase separation, thus yielding alumina monoliths with increased porosity and a hierarchical pore structure [64]. Further improvements in terms of pore size control and increasing porosities along with a greener route have been published recently by our group [68,91]. In this sub-section, we present an overview of the new possibilities offered by using di- and tricarboxylic acids as porogenes, as well as some new results concerning the influence of the gel-ageing temperature. Other parameters influencing the final α-Al_2_O_3_ pore structure include concentrations and solvents in the original reaction mixture, or the drying procedure. These are not included in our discussion.

As a starting point for our inquiries, we chose the synthesis presented by Baumann et al. [63], designated SG-Ref0, and its extension to hierarchical alumina materials via phase separation [64], designated SG-Ref100. Both samples were calcined at 1200 °C for 6 h to yield pure α-Al_2_O_3_, as shown in Figure 4. This procedure is applied to all samples in this section, unless otherwise indicated. The respective porosities are illustrated by SEM images and mercury intrusion histograms in Figure 5. The phase separation effects become obvious: the pore volume is more than tripled by generation of a secondary pore domain around 1300 nm. A hierarchical structure forms, granting access to the smaller pore domain centered around 130 nm.

In an effort to decrease costs, broaden the attainable range of pore sizes, and render the synthesis route more environmentally friendly, we replaced the phase separating agent PEO by di- and tricarboxylic acids, as published recently. We observed a change in phase separation regime from an entropy-driven mechanism (with PEO) to an enthalpy-driven one (with carboxylic acids). For a detailed study of this phenomenon, please refer to [68].

The use of di- and tricarboxylic acids also enables control over a broader range of pore sizes, from ≈ 110 nm up to several µm. Exemplary pore diameters are listed in Table 4. Selected SEM images in Figure 6 illustrate the structure directing properties of different additives. (For these new porogenes, the molar ratio of Al^3+^/additive is henceforth designated as *φ_Al_*). While dicarboxylic acids with longer carbon chains, such as adipic (C6) or glutaric acid (C5), have almost no influence on the α-Al_2_O_3_ microstructure, this effect becomes very strong for malonic (C3) and oxalic acid (C2). Pore diameters increase due to enlarged primary particles, which are formed by dicarboxylate-Al(III)-oligomers. Eventually, when doubling the amount of oxalic acid to attain *φ_Al_* = 5, phase separation occurs, yielding a bimodal pore system. Smaller macropores of *d_p_* ≈ 150 nm result from the sol-gel process itself, while a secondary pore domain with larger macropores ≥ 1 µm emerges between the spherical super-particles that form during phase separation.

Likewise, citric acid can be used to first enlarge primary pore diameters, and to induce phase separation when employed in sufficient amounts. Since it possesses an additional carboxylic acid functional group, one molecule of citric acid can link three instead of two Al(III)-nuclei. The amount of CA needed is hence smaller than the amount of a dicarboxylic acid necessary for a comparable result, regarding porosity values. However, pore sizes can be adjusted far better with dicarboxylic acids, with much narrower pore size distributions. The reason thereof can be found in the mechanism, proceeding via dicarboxylate-Al(III)-oligomers. Branched molecules like citric acid form larger but also more irregular primary particles than linearly composed dicarboxylic acids. Although this mechanism hypothesis cannot be proven with absolute certainty, it is strongly supported by several analyses [68,91]. The most definite and significant results are found in SEM imaging in Figure 6. Comparison of samples C3 (*φ_Al_* = 10) and CA34 (*φ_Al_* = 20) reveals the impact on the resulting *A_BET_*. For a slightly smaller *V_p_* (0.75 vs. 0.79 cm^3^/g), sample C3 exhibits an *A_BET_* of 10 m^2^/g due to its smaller *d_p_* of only 259 nm, while for sample CA34, larger primary particles resulting in larger pores (*d_p_* = 916 nm) lead to a reduced *A_BET_* of 5 m^2^/g.

Two samples listed in Table 4 are presented here for the first time to exemplarily illustrate the influence of the gel ageing temperature. By raising the same from 40 to 80 °C, we were able to significantly increase the porosity of sample SG-Ref0 by 200%, with a pore volume of 0.34 cm^3^/g and now two pore domains centered at 151 and 332 nm after calcination at 1200 °C, i.e., in α-Al_2_O_3_. The perfect bimodal pore structure of sample SG-Ref0-(80), depicted by the red mercury intrusion curves in Figure 7, is not a result of phase separation, however, but arises from accelerated Ostwald ripening at the elevated ageing temperature of 80 °C. This promotes the formation of larger particles along with the smaller ones that can already be observed at lower ageing temperatures. The other two intrusion curves in Figure 7 support this hypothesis. There are distinct mesopore domains, centered at 8 nm after calcination at 600 °C, or 15 nm after calcination at 950 °C, respectively. The macropores then display a broad distribution of pore sizes ranging up to 300, or 600 nm, respectively. Due to sintering effects during calcination to α-Al_2_O_3_, the pore domains eventually form two separate modes.

Interestingly, malonic acid seems to suppress the development of a bimodal pore structure also at an elevated gel ageing temperature, as can be observed for sample SG-C3-(80), aged at 80 °C. Mercury intrusion revealed perfectly monomodal pore size distributions for both malonic acid samples, centered at modal pore diameters of 259 nm for SG-C3 (aged at 40 °C), or 786 nm for SG-C3-(80), respectively. This is yet another piece of evidence supporting our hypothesis of dicarboxylate-Al(III)-oligomers. These intermediate species act conjointly with the elevated ageing temperature, *ergo* increased hydrolysis rate, to favor the formation of larger particles, leaving no smaller primary particles to develop a second pore domain in the small mesopore range.

In summary, the epoxide-mediated sol-gel process starting from AlCl_3_·6H_2_O offers many ways of controlling the pore diameter and pore volume in the resulting α-Al_2_O_3_ material. The prevailing method of determining these values is mercury intrusion, due to its range of analysis and the nature of α-Al_2_O_3_. While pore volumes can be boosted up to 1.4 cm^3^/g (and possibly even higher), pore diameters attain values of 110 nm and larger, despite all efforts presented here in reducing the same. This implies that for maximizing *A_BET_*, we are above all restricted by the lower pore diameter limit, as calculations in Section 4.1 demonstrate.

### 5.2. Mutual Cross-Hydrolysis in Combined Sol-Gel Synthesis

The route employed for the mutual cross-hydrolysis has been briefly reported as a means to synthesize yttria–alumina [176]. While several other publications also report syntheses starting from these two types of precursors, at least one of the two is peptized or hydrolyzed prior to combining both precursors [163,164,182,183,184,185,186].

By combining an aluminum alkoxide and an aluminum salt, we disclose at this point a novel way of preparing porous alumina gels. Upon dissolution, the aluminum salt dissociates into solvated anions and [Al(H_2_O)_6_]^3+^-complexes. In a concerted, simultaneous cross-hydrolysis, the Al-hexaaqua-complexes are then hydrolyzed by the much more reactive alkoxide, while at the same time the alkoxide is also hydrolyzed in the strongly acidic medium by nucleophilic substitutions of protonated –OR-groups by water. It hence dispropotionates into [Al(OH)_n_(H_2_O)_6−n_]^(3−n)^-complexes and solvent molecules (isopropanol). This holds also true for assays designated as “water-free”, since the crystal water of the aluminum salt is still present. The Al-complexes undergo hydrolytic oxolation, yielding an alumina network quite similar to the one obtained during the classic sol-gel process. While it might seem redundant to use two different precursors for the same metal, the great benefit arising from this approach is that this novel synthesis route does not necessitate the use of a carcinogenic epoxide.

In this section, we discuss two representative samples prepared via the novel mutual cross-hydrolysis approach. MCH-w samples were prepared with equimolar amounts of aluminum salt AlCl_3_·6H_2_O) and aluminum alkoxide (AIP) in a solvent mixture of water and ethanol at 80 °C in a laboratory furnace. Mercury intrusion data in Figure 8 illustrate the evolution of the pore system for MCH-w samples calcined at 600 °C or 1200 °C, to yield amorphous or α-Al_2_O_3_, respectively, as shown in Figure 9.

In a “water-free” assay, MCH-o samples were prepared with three parts of aluminum alkoxide (ASB) and two equivalents of aluminum salt Al(NO_3_)_3_·9H_2_O in a solvent mixture of iso-propanol and ethanol, hence containing only the crystal water of the Al(NO_3_)_3_·9H_2_O. MCH-o samples were also placed in a laboratory furnace at 80 °C.

The expected effect of elevated calcination temperatures on the pores is similar for both MCH assays, and comparable to the epoxide-mediated route. When attaining the α-modification at 1200 °C, pore diameters grow to values > 100 nm, since smaller pores collapse to merge with the larger ones due to sintering effects [187,188]. This process can be regarded as a temperature-promoted Ostwald ripening [189].

The pore volume *V_p_* remains virtually unchanged for MCH-w samples, dropping from 0.47 to 0.42 cm^3^/g. With approximately 0.21 cm^3^/g, half of the total pore volume *V_p_* can be assigned to the *d_p,mod_* of 160 nm for MCH-w. This is twice as much as for sample SG-Ref0, which exhibits a *V_p_* of 0.10 cm^3^/g for the *d_p,mod_* of 116 nm. When carried out as a “water-free” synthesis in organic solvents, pore size distributions become narrower but at the expense of diminishing pore volumes (cf. Figure 8). For a standard MCH-o sample, *V_p_* decreases from initially 0.76 to 0.22 cm^3^/g, with *d_p,mod_* increased to 130 nm, still with an extremely narrow pore size distribution. While this also marks an improvement compared to the epoxide-mediated SG-Ref0, attempts to increase the pore volume by additives were not successful to date.

SEM images in Figure 10 illustrate the α-Al_2_O_3_ porosity obtained by the novel route in comparison with the additive-free SG-Ref0-(80) (aged at 80 °C, calcined at 1200 °C). Most importantly, this novel route replaces the carcinogenic epoxide by a proton scavenger that not only decomposes into harmless solvent molecules but additionally contributes to the formation of the desired alumina network.

Ongoing studies in our laboratories are treating the influence of several reaction parameters that have hitherto not been investigated. A follow-up publication is in preparation.

### 5.3. Manganese-Assisted α-Transition in AAO Membranes

The results presented in this sub-section are excerpted from a publication, which is currently being prepared in our group [168]. Evidently, those results do not claim to be comprehensive but merely represent a preview of the full article, to be published in the near future.

Combination of Tsyrulnikov’s solid solutions concept [169] with highly ordered porous membranes prepared via anodic oxidation of aluminum chips (AAO membranes) yields thermally and chemically stable membranes consisting of XRD-pure α-Al_2_O_3_, with preservation of the delicate hexagonal pore system. However, to this end, impregnation of AAO membranes with a manganiferous solution is imperative prior to calcination.

SEM images in Figure 11 show the tremendous effect of this treatment regarding the preservation of the pore system. Calcination of pure AAO membranes yields α-Al_2_O_3_ only at 1100 °C, at the expense of a fatal deterioration of the membrane structure and its overall intactness. By impregnation with the manganiferous solution, the α-transition temperature can be lowered by about 200 °C. Figure 12 presents a scheme suggested by Tsyrulnikov et al. [169] to explain the effect added Mn has on the phase transition in the alumina system. Their concept is based on the idea of a solid solution of Mn(III)-ions within the alumina lattice, wherein Mn-doped α-Al_2_O_3_ already appears at 900 °C. This reduction of the α-transition temperature enables complete preservation of the delicate pore structure.

Image analysis of the obtained membranes showed a remarkable regularity of the pore system (cf. Figure 13). Hexagonal pore centers exhibit almost perfect inter-pore angles of 60°, which corresponds to the optimum distribution in a closest packing. Likewise, the regularity of the respective inter-pore distances, with a first neighbor distance of 100 nm, corresponds to the optimum distribution. Only when looking at the long-range order, some irregularities become apparent. We observe a polycrystalline array of intrinsically very homogeneous pore domains, at the boundaries of which defects occur due to differing orientation of the individual crystallites. Pore sizes as analyzed from SEM images are listed in Table 5, illustrating the extremely narrow pore size distribution that can be obtained with the aid of Mn-assisted phase transformation in AAO membranes.

This pore diameter of 44 nm (with an extremely small deviation of approximately ±2 nm) is the smallest one published for pure α-Al_2_O_3_ published hitherto. We hence included this finding in our calculations of the theoretically attainable porosities and corresponding *A_BET_* of porous α-Al_2_O_3_ materials in Section 4.1. There are, however, two major restraints regarding this issue. Firstly, comparing pore diameters calculated by different methods—in this case SEM image analysis vs. mercury intrusion—is not entirely accurate. More importantly, the astonishingly small pore diameter of 44 nm was only determined on the surface of a highly organized α-Al_2_O_3_ material, with information lacking about the bulk phase. Since AAO membranes do not withstand the pressure necessary for mercury intrusion analysis, we cannot provide reliable values thereof. This value is hence not a sound basis for calculating theoretically attainable specific surface areas in materials synthesized via a completely different approach.

### 5.4. Application of Solid Solutions Concept to Sol-Gel Materials

Having proven the above described solid solutions concept, we strove to transfer the Mn-assisted α-transition to classic sol-gel materials. Samples were prepared according to the modified standard synthesis described in Section 3.1.1, and impregnated with 1 M aqueous solution of Mn(NO_3_)_2_ or Fe(NO_3_)_3_, as described in Section 3.1.4, prior to calcination at temperatures ranging from 900 to 1050 °C. For comparison, blank sol-gel samples and sol-gel samples prepared from 95 mol-% Al- and 5 mol-% Mn- or Fe-precursors, respectively, were also subjected to the same calcination program.

In Figure 14, XRD patterns of blank and impregnated samples are shown, both for the sol-gel and the AAO approach. Both Mn-impregnated AAO membranes and sol-gel samples can be converted into α-Al_2_O_3_ at temperatures as low as 900 °C (Figure 14A,B). While for the AAO membrane, γ-Al_2_O_3_ reflexes are still present, the sol-gel sample SG-Mn-imp-900-6 contains a considerable amount of hausmannite Mn_3_O_4_. Nonetheless, its *A_BET_* amounts to 23 m^2^/g, which is a remarkably high value for an α-Al_2_O_3_ sample without any transition alumina; the only moderately crystalline γ-Al_2_O_3_ reference SG-Ref-900 exhibts an *A_BET_* of 152 m^2^/g. Despite this expectable shortfall, the specific surface area of the Mn-impregnated sol-gel sample is still 64% higher than the 14 m^2^/g obtained for the Mn-impregnated AAO sample (A), which also still contains γ-Al_2_O_3_.

After intensifying the calcination conditions to 950 °C for 168 h, comparative XRD patterns in Figure 15 show no significant improvement in the degree of α-conversion of the impregnated sol-gel material, with hausmannite Mn_3_O_4_ still present (Figure 15A, sample SG-Mn-imp-950-168). Interestingly, the Mn-doped sol-gel sample SG-Mn05-950-168 (synthesized from 95 mol-% AlCl_3_ and 5 mol-% MnCl_2_) does not show any tendency of forming α-Al_2_O_3_ at 950 °C (B) but exhibits almost the same reflection pattern of a poorly crystallized θ-Al_2_O_3_ as the undoped reference sample SG-Ref0-950-168 (C). The Mn-ions appear to be immobilized within the alumina matrix, as they are undetectable by XRD, rendering them unable to facilitate an earlier α-transition. This is mainly due to the crystal structure of manganese oxides. While many transition metal oxides adopt the hexagonal corundum structure, both manganese oxides expected at 950 °C (bixbyite Mn_2_O_3_ and hausmannite Mn_3_O_4_) crystallize in a defect spinel structure, which is closely related to the structure of γ-Al_2_O_3_.

By raising the calcination temperature to 1050 °C, however, a similarly pure α-Al_2_O_3_ can be obtained by doping (sample SG-Mn05-1050-12) as for impregnated alumina SG-Mn-imp-1050-12, containing reduced amounts of hausmannite, next to the α-Al_2_O_3_ (cf. Figure 16B,C). Both samples differ significantly from the reference sample SG-Ref0-1050-12, which shows a poorly crystallized mixture of γ-Al_2_O_3_ and θ-Al_2_O_3_ (cf. Figure 16F).

Attempts to rid converted α-Al_2_O_3_ of remaining manganiferous species were hence carried out on the impregnated sample SG-Mn-imp-1050-12. After impregnation and calcination, its Mn content amounted to 7.7 wt.%, as determined by ICP-OES, or 8.6 wt.% (SEM-EDX). This share could be reduced to ≈ 1.4 wt.% by acid leaching. Results were virtually identical, whether leaching was carried out in concentrated HCl at ambient temperature, or in aqua regia under reflux for 4 h. In view of the applied conditions, further reduction is highly unlikely. We hence conclude that after Mn-assisted α-conversion, ≈ 0.5 mol.-% of Mn remain within the α-Al_2_O_3_. Since this value is below the detection limit of X-ray diffraction, the corresponding pattern of sample SG-Mn-ex, given in Figure 16A, shows pure α-Al_2_O_3_. Its *A_BET_* amounts to 10 m^2^/g, which is the same value as for sample SG-C3, synthesized using the same amount of the same additive (oxalic acid, *φ_Al_* = 10) but converted into α-Al_2_O_3_ by standard calcination at 1200 °C for 6 h. Again, the considerably higher *A_BET_* of 82 m^2^/g for the undoped reference, which exhibits no α-Al_2_O_3_ reflexes, is in good agreement with our expectations.

For Fe-doping, an effect contrary to Mn-doping can be observed. Hematite α-Fe_2_O_3_ crystallizes in the hexagonal corundum structure [170]. When introduced into sol-gel alumina via Fe-precursors, the dopant Fe(III)-ions direct the surrounding alumina network into the (thermodynamically favorable) corundum structure [171]. This enables a facilitated α-transition at temperatures lowered by more than 100 °C, as the XRD pattern in Figure 16D illustrates, yet at the expense of losing much of the specific surface area: Sample SG-Fe05-1050-12 exhibits an *A_BET_* of only 2 m^2^/g. Impregnation with a ferreous precursor solution does not yield the same effect as with Mn, although the α-transition is also facilitated by the only surficial presence of Fe(III)-ions (sample SG-Fe-imp-1050-12, Figure 16E). This enhancing effect of Mn can be explained by the formation of solid solutions of highly mobile Mn-ions on the alumina surface [169]. Although Fe has also been reported to form solid solutions with corundum [172], this effect apparently is not as potent for superficially distributed Fe.

In conclusion, an even distribution throughout the whole sample implies a low ion mobility for dopants. Due to the crystal structures of their respective oxides, this immobility favors the formation of α-Al_2_O_3_ in the case of Fe-doping but does not have any effect for Mn-doping. α-Fe_2_O_3_ will impose the corundum structure upon the transition alumina, while manganese oxides retain the (defect) spinel structure. In contrast, impregnation with manganiferous or ferreous solution results in only superficial distribution of the dopant ions, along with a high mobility. This enables Mn(III)-ions to form the solid solutions described above (Figure 12) and thus favor the formation of α-Al_2_O_3_, while Fe(III)-ions are trapped on the surface and can hence not impose the corundum structure onto the transition alumina with its defect spinel structure.

Moreover, 84% of manganiferous species can be extracted after α-conversion by acid leaching in aqua regia under reflux. This results in an essentially pure α-Al_2_O_3_ containing only 0.5 mol.-% of Mn, with a significantly higher specific surface area of 23 m^2^/g, which is more than double the *A_BET_* obtained after standard calcination at 1200 °C.

### 5.5. Pore Protection by Carbon Filling

The meso- and macropores of θ-Al_2_O_3_ materials were filled with carbon with the intent of preventing their collapse during the θ→α-transition, which takes place at temperatures around 1200 °C. To this end, two different kinds of θ-Al_2_O_3_ samples were impregnated with near-saturated sucrose solution, which was then carbonized under nitrogen flow. These two steps were repeated until mass constancy, prior to calcination in vacuo.

*NB*: A few attempts were also made using a nitrogen gas flow during calcination at 1250 °C, yielding a mixture of γ-Al_2_O_3_ and aluminum nitride (XRD not shown). This result leads us to contest any publications stating conversion of γ- into α-Al_2_O_3_ under such conditions.

For impregnated samples, carbon content ranged from 34%–43% after the final carbonization step, which is significantly higher than the 7% attained by Murrell et al. in their 1978 patent [46]. The degree of conversion into α-Al_2_O_3_, which is decisive for the assessment of this route, was monitored by X-ray diffraction. The corresponding diffraction patterns are shown in Figure 17, with in vacuo calcination conditions indicated. It becomes apparent that a complete θ→α-transition is difficult to attain by this pore-protection approach. Both industrial pellets and sol-gel alumina are only partially converted into α-Al_2_O_3_ even after calcination at 1250 °C for 12 h followed by a second calcination step at 1350 °C for 2 h, with significant amounts of θ-Al_2_O_3_ still remaining. Sol-gel (SG-pp) samples appear to be less inclined to undergo α-transition than extruded commercial pellets, presumably due to less residual surficial OH-groups and a better pre-organization in the latter ones. Specific surface areas amount to 54 m^2^/g for industrial pellets, and 64 m^2^/g for sol-gel alumina, respectively, which also indicates a poorer degree of α-conversion for the sol-gel samples.

Interestingly, even doping with 10 wt.% α-Al_2_O_3_ seeds cannot induce complete θ→α-transition in sol-gel alumina samples (designated α* in Figure 17). Moreover, their specific surface areas seem to be irrespective of the calcination conditions, yielding of 76–81 m^2^/g. However, for these samples, the second calcination step at 1350 °C was omitted, since α-seeding had previously been shown to reduce the θ→α-transition temperature to 1100 °C for calcination in air of porous samples without carbon filling.

Considering all results presented above, we conclude that carbon is indeed an apt material for protecting pores in transition alumina, as numerous publications and patents state, thus generating HSSA alumina. However, this preservation of (meso) porosity happens at the expense of an incomplete conversion into α-Al_2_O_3_ by preserving the transition alumina crystal structure.

Moreover, upon a second calcination after carbon removal, these materials are then eventually converted into α-Al_2_O_3_, as shown by the corresponding XRD pattern in Figure 17. At the same time the beforehand increased specific surface area drops to about 5 m^2^/g.

We thus believe to have provided strongly supportive arguments of our reasonable doubts regarding the feasibility of completely transforming pore-protected transition alumina into the α-phase. Our findings second the ones reported by Schüth et al. [57], stating that pure α-Al_2_O_3_ can hardly be obtained via this route.

## 6. Conclusions

Several different routes supposedly yielding macroporous α-Al_2_O_3_ were investigated both by assessment of available literature and experimental approaches. Regarding the general possibility of obtaining high specific surface area α-Al_2_O_3_, literature data on thermodynamics suggest that for *A_BET_* > 175 m^2^/g, γ-Al_2_O_3_ might in fact be the thermodynamically stable modification of Al_2_O_3_. Investigation of the respective crystallite sizes shows that one boehmite crystallite subdivides into several γ-Al_2_O_3_ crystallites, yet the eventual α-Al_2_O_3_ crystallite resumes the dimensions of the original boehmite grain. There is hence also a considerable kinetic barrier for the conversion of transition alumina into corundum.

Keeping in mind these difficulties, the diversity of methods used to impose porosity on this intrinsically non-porous material is remarkable. Best results in terms of maximizing the specific surface area were reported by employing sol-gel methods (*A_BET_* 16–24 m^2^/g). This also applies to our own experimental results of sol-gel syntheses with simple di- and tricarboxylic acids as novel porogenes (*A_BET_* of up to 12 m^2^/g). Moreover, we report on the novel concept of mutual cross-hydrolysis, combining an aluminum salt and an alkoxide to obtain porous gels without the use of cancerogenic epoxides. Controlling the pore size and pore volume via this synthesis route is part of ongoing research and will be published separately.

Despite several patents on preservation of high specific surface areas by carbon-filling of pores, most of the reported processes raise questions regarding the complete α-transition, or the authors even proceed to qualify their own results. Our experimental findings confirm these difficulties, arising from kinetic barriers in terms of differences in free surface energy and crystallite size incongruence, manifested by blocked pores. These impair the formation of α-Al_2_O_3_ seeds from much smaller γ-Al_2_O_3_ crystallites with lower free surface energy.

This kinetic barrier can be abolished by avoiding the corundum formation sequence proceeding via γ-Al_2_O_3_, i.e., preparing α-Al_2_O_3_ from diaspore α-AlO(OH). Thus, it is undoubtedly possible to generate α-Al_2_O_3_ with an *A_BET_* as large as 160 m^2^/g but most likely not exceeding 175 m^2^/g, as thermodynamic calculations suggest. Despite the still very high specific surface area attainable in diaspore-derived corundum, there are two major drawbacks. Firstly, the preparation process of diaspore is extremely energy intensive and almost always necessitates diaspore seeding crystals, and secondly, the elevated specific surface area drastically drops when low-temperature corundum is heated above 600–800 °C.

Another way of obtaining alumina with enhanced specific surface areas at elevated temperatures consists of adding certain dopant ions. While rare earths tend to stabilize γ-Al_2_O_3_ even at temperatures > 1000 °C, manganese has a different effect on the γ→α-transition. When introduced into transition alumina, Mn forms solid solutions of Al(III)-ions within hausmannite and, more importantly, of Mn(III)-ions within corundum at 900 °C. This permits the obtainment of α-Al_2_O_3_ at a relatively low temperature, implying the preservation of porosity to a much larger extent. Subsequently, all manganiferous species can be extracted by acid leaching, yielding pure porous α-Al_2_O_3_. We demonstrated that this mechanism does not only work for anodically oxidized alumina membranes but also for highly porous sol-gel alumina, hence enabling us to preserve a considerable part of the previously generated porosity during the α-transition. By Mn-impregnation, we obtained a remarkable *A_BET_* of 23 m^2^/g for a sol-gel derived 99.5% pure α-Al_2_O_3_ at 900 °C, which is among the highest values reported to date.

In conclusion, the synthesis of porous α-Al_2_O_3_ remains a challenging field of research. Due to the strict thermodynamic constraints, advances in increasing the specific surface area move slowly. While certain strategies, such as pore protection by carbon-filling, have proven to be unfeasible due to kinetic barriers, sol-gel routes, possibly in combination with a Mn-assisted α-transition, look promising. Further efforts will be needed from the scientific community to achieve the goal of a temperature-stable corundum with high specific surface area.

## Figures and Tables

**Figure 1 materials-13-01787-f001:**
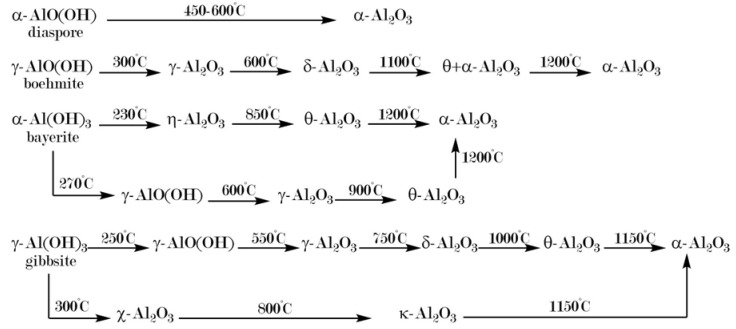
Phase transformations in the alumina system according to [9].

**Figure 2 materials-13-01787-f002:**
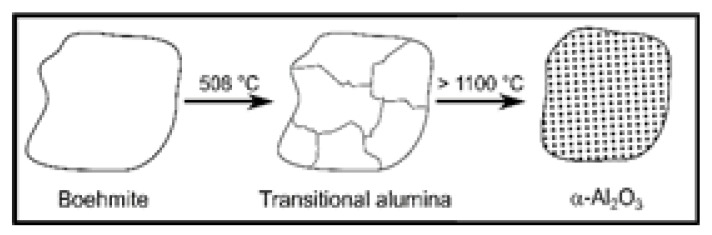
Correlation between boehmite and corundum crystallite sizes adapted from [11].

**Figure 3 materials-13-01787-f003:**
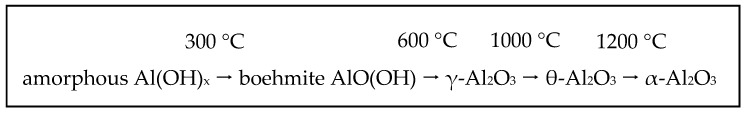
Phase transformations following the boehmite path, adapted from [68].

**Figure 4 materials-13-01787-f004:**
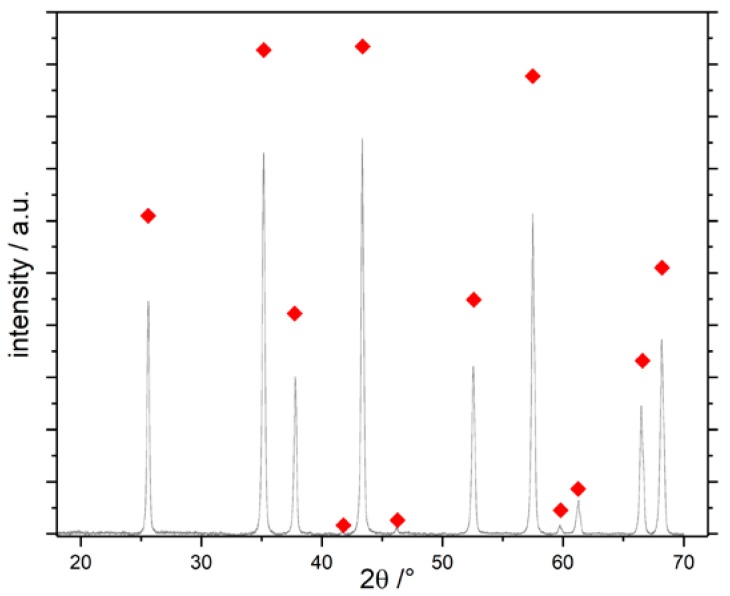
XRD pattern of sample SG-Ref0 after calcination at 1200 °C for 6 h confirms complete transformation to α-Al_2_O_3_ (
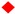
). All other samples display the same diffraction pattern.

**Figure 5 materials-13-01787-f005:**
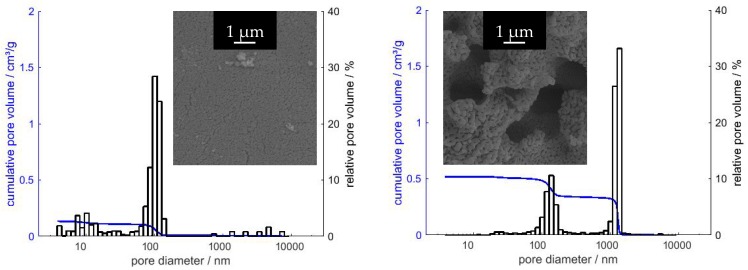
Mercury instruction histograms and SEM images of the starting point syntheses SG-Ref0 (left, without polyethylene oxide [PEO]), and SG-Ref100 (right, with addition of PEO of Mm 900,000). Both samples were calcined at 1200 °C for 6 h.

**Figure 6 materials-13-01787-f006:**
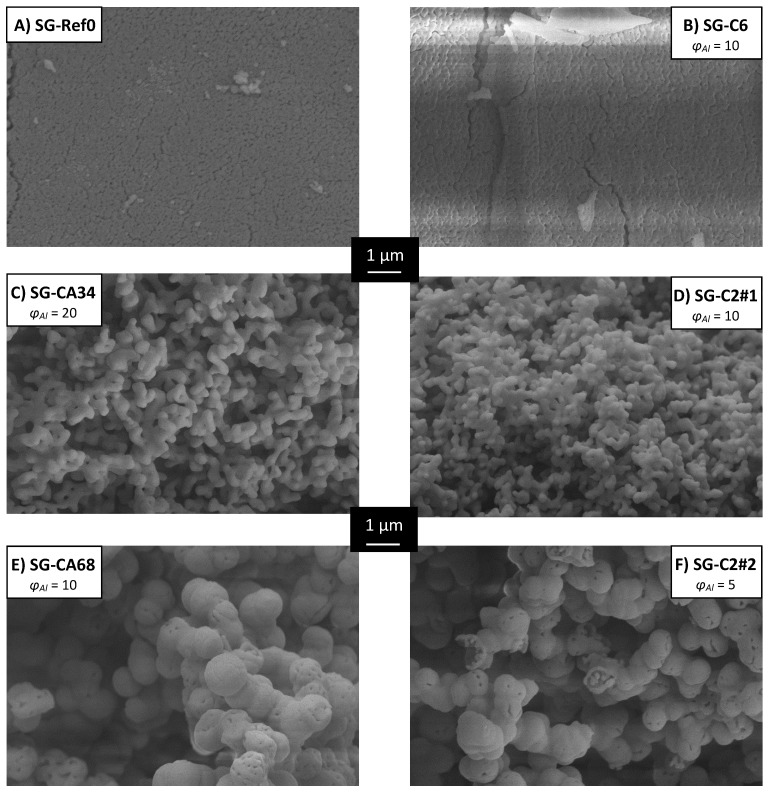
SEM images of sol-gel samples reveal the influence of different additives (**A**) SG-Ref0: none, (**B**) SG-C6: adipic acid, (**C**) SG-CA34 and (**E**) SG-CA68: citric acid, (**D**) SG-C2#1 and (**F**) SG-C2#2: oxalic acid) and their respective amount on the microstructure. *φ_Al_* designates the molar ratio of Al^3+^/additive. All samples were calcined at 1200 °C for 6 h to yield α-Al_2_O_3_ (cf. Figure 4) [91].

**Figure 7 materials-13-01787-f007:**
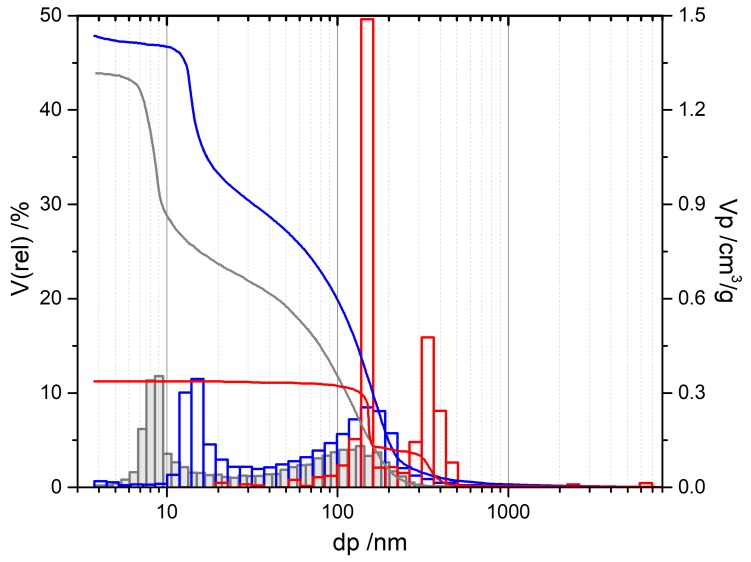
Superimposed mercury intrusion data of sample SG-Ref0-(80), aged at 80 °C, and calcined at 600 °C (gray), 950 °C (blue), and 1200 °C (red).

**Figure 8 materials-13-01787-f008:**
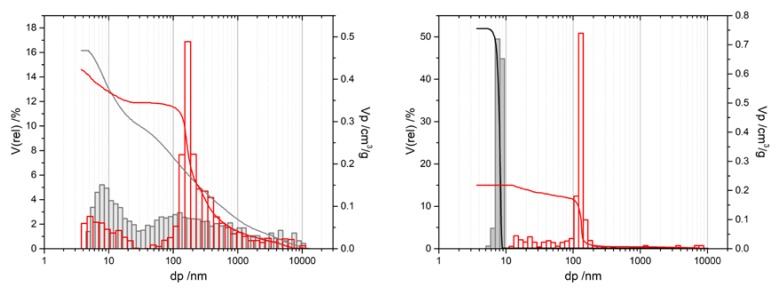
Superimposed mercury intrusion data of samples MCH-w (**left**) and MCH-o (**right**), calcined at 600 °C (gray) and 1200 °C (red), respectively. NB: For better legibility, scales are adjusted to the requirements of the data.

**Figure 9 materials-13-01787-f009:**
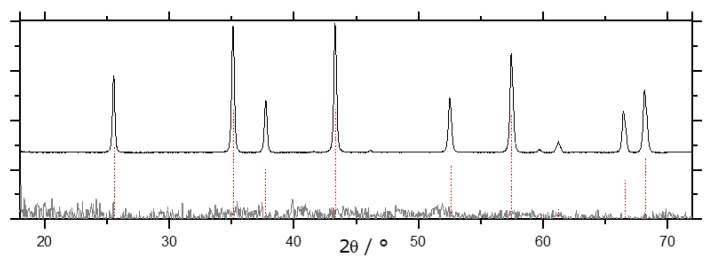
XRD patterns of sample MCH-w, calcined at 600 °C (gray) and 1200 °C (black), respectively. Red dotted lines indicate α-Al_2_O_3_ reflexes.

**Figure 10 materials-13-01787-f010:**
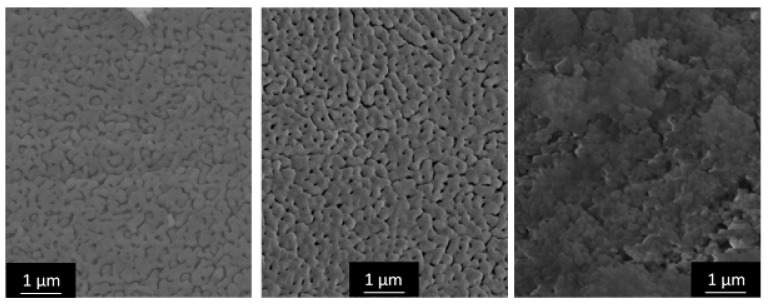
Comparison of SEM images of samples MCH-w (left), MCH-o (center), and SG-Ref0-(80) (right, epoxide-mediated), all calcined at 1200 °C, illustrate the increased pore volume obtained via mutual cross-hydrolysis.

**Figure 11 materials-13-01787-f011:**
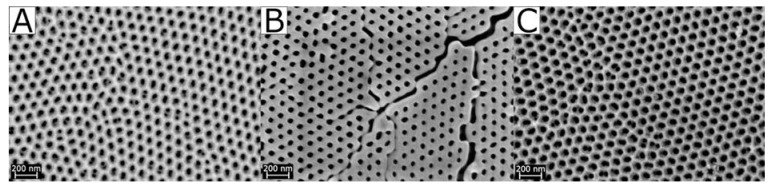
SEM images of anodized aluminum oxide (AAO) membranes having undergone different treatment: (**A**) AAO-0 pristine, (**B**) AAO-1100 after calcination at 1100 °C, and (**C**) AAO-900-Mn calcined at 900 °C after impregnation with a manganiferous solution [168].

**Figure 12 materials-13-01787-f012:**
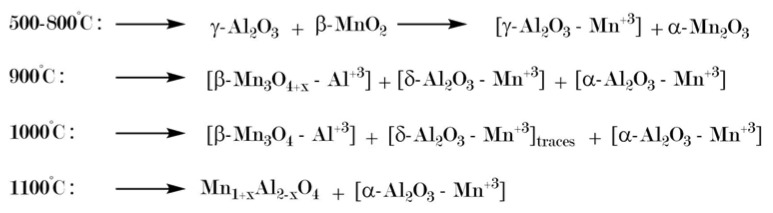
Suggested mechanism of phase transitions in the Mn-doped alumina system, adapted from [169].

**Figure 13 materials-13-01787-f013:**
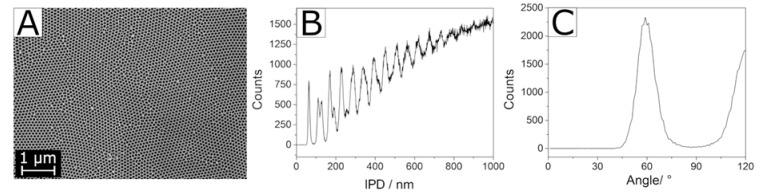
Automatized image analysis of SEM images (**A**) of porous AAO membranes, transformed into α-Al_2_O_3_ by Mn-impregnation, with Pore Distribution Function (**B**) and Angle Distribution Function (**C**) [168].

**Figure 14 materials-13-01787-f014:**
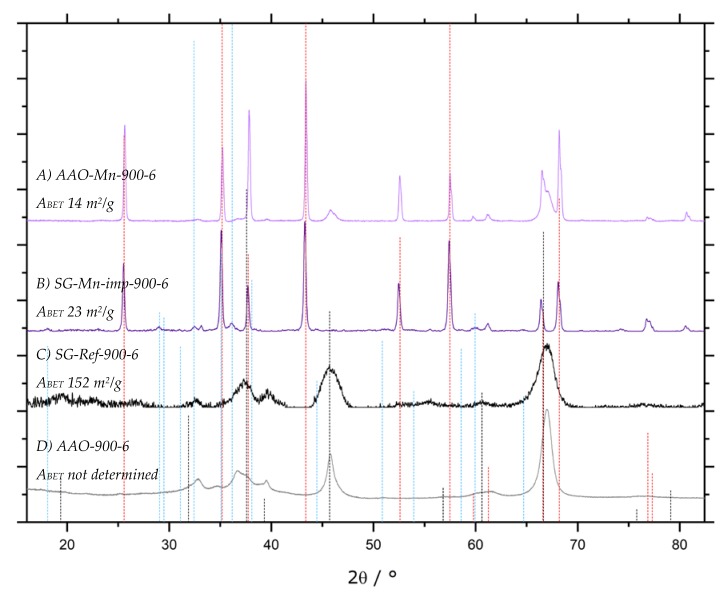
XRD patterns of AAO (**A**,**D**) and sol-gel samples (**B**,**C**) calcined at 900 °C for 6 h. Impregnation with manganiferous precursor solution yields almost pure α-Al_2_O_3_ for AAO samples (**A**), with some γ-Al_2_O_3_. Mn-impregnated sol-gel material (**B**) shows only α-Al_2_O_3_ reflexes, along with some hausmannite. Patterns (**C**,**D**) arise from the reference samples without impregnation. Red dotted lines indicate α-Al_2_O_3_, black dotted lines indicate γ-Al_2_O_3_, hausmannite reflexes are marked in blue.

**Figure 15 materials-13-01787-f015:**
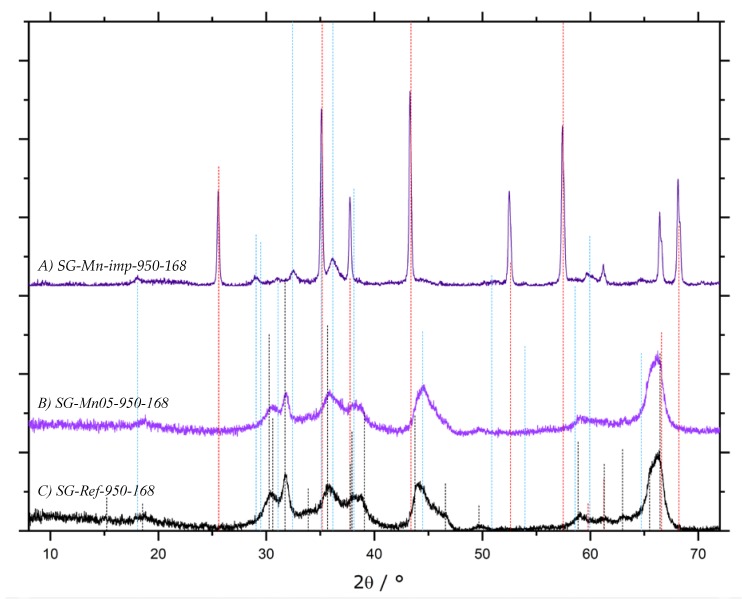
XRD patterns of sol-gel samples calcined at 950 °C for 168 h. Sample SG-Mn-imp950-168 (**A**), impregnated with manganiferous precursor exhibits all reflexes of α-Al_2_O_3_, along with hausmannite Mn_3_O_4_, and some remaining θ-Al_2_O_3_, while the pure alumina sample SG-Ref0-950-168 (**C**) shows a pattern of a poorly crystallized θ-Al_2_O_3_. The same holds true for sample SG-Mn05-950-168 (**B**), synthesized from 95 mol-% Al- and 5 mol-% Mn-precursors. Red dotted lines indicate α-Al_2_O_3_, black dotted lines indicate θ-Al_2_O_3,_ hausmannite reflexes are marked in blue.

**Figure 16 materials-13-01787-f016:**
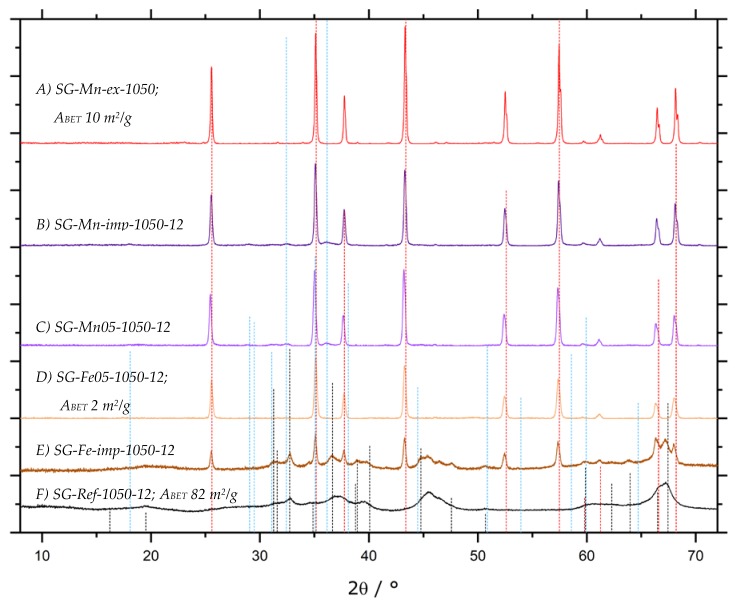
XRD patterns of sol-gel samples calcined at 1050 °C for 12 h. The undoped reference sample SG-Ref0-1050-12 (**F**) shows a pattern of poorly crystallized θ-Al_2_O_3_. Impregnation with ferreous precursor solution (**E**) has a much weaker effect than the one observed for Mn (**B**), which contains only α-Al_2_O_3_ and hausmannite Mn_3_O_4_. Much of the hausmannite can be dissolved by acid leaching, giving an XRD-pure α-Al_2_O_3_ pattern (**A**). Sample SG-Mn05-1050-12 (**C**), synthesized from 95 mol-% Al- and 5 mol-% Mn-precursors, gives a similar pattern to the impregnated sampe (**B**), while incorporation of hexagonal α-Fe_2_O_3_ from ferreous precursors enables a facilitated α-transition (**D**, sample SG-Fe05-1050-12), with no detectable ferreous crystal phases. Red dotted lines indicate α-Al_2_O_3_ reflexes, black dotted lines indicate θ-Al_2_O_3_ reflexes. Hausmannite reflexes are marked in blue.

**Figure 17 materials-13-01787-f017:**
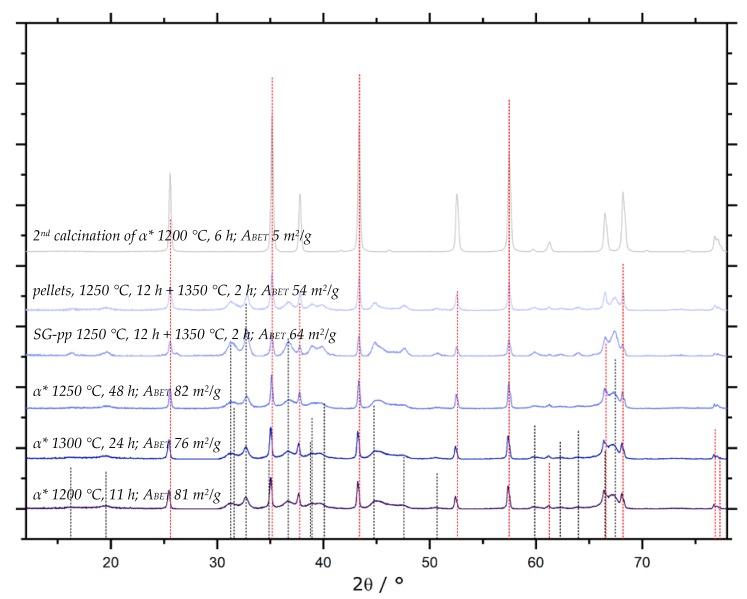
XRD patterns of pore-protected θ-alumina samples with indicated calcination conditions and A_BET_ determined from N_2_ sorption measurements. The three bottom patterns arise from α-doped sol-gel alumina (α*). Red dotted lines indicate α-Al_2_O_3_ reflexes, black dotted lines indicate θ-Al_2_O_3_ reflexes.

**Table 1 materials-13-01787-t001:** Enthalpy of formation (Δ*H_f_*) and enthalpy of transition to α-Al_2_O_3_ (Δ*H_→α_*), for different cubic alumina modification, as compiled from [2,3].

Alumina Modification	Δ*H_f_*/kJ/mol [2]	Δ*H_→α_*/kJ/mol [2]	Δ*H_→α_*/kJ/mol [3]
γ-Al_2_O_3_	−1656.9	−18.8	−22.2
κ-Al_2_O_3_	−1662.3	−13.4	−15.1
δ-Al_2_O_3_	−1666.5	−9.2	−11.3
α-Al_2_O_3_	−1675.7	n/a	n/a

**Table 2 materials-13-01787-t002:** List of all samples discussed in the calculation and results sections of this review article.

Exp. Section	Preparation Method	Sample Name	Short Description	Discussed in Sections	Ref.
3.1.1	epoxide-mediated sol-gel	SG-Ref0	without additive	4.1, 5.1, 5.2	[68,91]
		SG-Ref0-(80)	SG-Ref0, aged at 80 °C	5.1, 5.2	n/a
		SG-Ref100	with PEO 900,000	4.1, 5.1	[68]
		SG-CA*xx*	with 0.*xx* g citric acid	4.1, 5.1	[68]
		SG-C2#1	with oxalic acid	5.1	[91]
		SG-C2#2	with oxalic acid (double)	4.1, 5.1	[91]
		SG-C3	with malonic acid	4.1, 5.1	[91]
		SG-C3-(80)	SG-C3, aged at 80 °C	5.1	n/a
		SG-C4	with succinic acid	4.1, 5.1	[91]
		SG-C5	with glutaric acid	5.1	[91]
		SG-C6	with adipic acid	5.1	[91]
3.1.2	sol-gel	MCH-w	mutual cross-hydrolysis + water	5.2	n/a
		MCH-o	mutual cross-hydrolysis in organic solvents (“water-free”)	5.2	n/a
3.1.3 3.1.4	anodic oxidation	AAO-0	pristine	5.3	[168]
		AAO-1100	calcined at 1100 °C	5.3	[168]
		AAO-Mn-900	impregnation with 1 M Mn(NO_3_)_2_, calcined at 900 °C	5.3, 5.4	[168]
3.1.4	calcination after impregnation	SG-Mn-imp	SG-Ref0, 1 M Mn(NO_3_)_2_	5.4	n/a
		SG-Fe-imp	SG-Ref0, 1 M Fe(NO_3_)_3_	5.4	n/a
		SG-Mn05	SG-Ref0, prepared with 5% MnCl_2_	5.4	n/a
		SG-Fe05	SG-Ref0, prepared with 5% FeCl_3_	5.4	n/a
	Mn extraction	SG-Mn-ex	SG-Mn-imp leached in HCl	5.4	n/a
3.1.5	carbon filling	SG-pp	θ-alumina from SG route, pore-protected with carbon	5.5	n/a
		α*	SG-pp, 10 wt.% α-Al_2_O_3_ particles	5.5	n/a
		pellets	pore-protected commercial θ-alumina pellets	5.5	n/a

**Table 3 materials-13-01787-t003:** Exemplary specific surface areas as a function of *d_p,mod_* and *V_p_*, as obtained by mercury intrusion, and corresponding *A_BET_* calculated from nitrogen sorption. Sample names are explained in Section 5.1 below. However, their signification is not relevant for this consideration. Samples appear in order of the deviation ratio |*A_BET_* − *A_theo_*|/*A_BET_*.

Sample Name	*d_p_*/nm	*V_p_*/cm^3^/g	*A_theo_*/m^2^/g	*A_BET_*/m^2^/g	|*A_BET_* − *A_theo_*| *A_BET_*
SG-C5	195	0.38	7.8	8	0.03
SG-Ref100	135; 1345	0.18; 0.34	6.3	6	0.05
SG-CA91	140; 5320	1.14	5.4	5	0.08
SG-CA27	326	0.67	8.2	9	0.09
SG-CA45	137; 2000	0.14; 0.90	6.3	7	0.10
SG-C3	259	0.75	11.6	10	0.16
SG-CA68	158; 3700	0.23; 1.09	7.0	6	0.17
SG-CA23	197	0.41	8.3	10	0.17
SG-Ref0	116	0.12	4.1	5	0.18
SG-C4	182	0.43	9.5	8	0.19
SG-CA14	150	0.17	4.5	6	0.25
SG-CA34	916	0.79	3.5	5	0.30
SG-C2#2	147; 2530	0.18; 1.35	7.0	5	0.40
SG-CA19	164	0.25	6.1	12	0.49
SG-C6	138	0.18	5.2	3	0.73
SG-C2#1	838	1.18	5.6	3	0.87

**Table 4 materials-13-01787-t004:** Pore diameters and volumes for different amounts of selected additives in the sol-gel synthesis of macroporous α-Al_2_O_3_ after calcination at 1200 °C. Most data are adapted from [68] and [91]. Samples SG-Ref0-(80) and SG-C3-(80) are presented here for the first time.

Sample	Additive	*φ_Al_* ^[a]^	*V_p_*^[b]^/cm^3^/g	*d_p_*^[b]^/nm	*A_BET_*^[c]^/m^2^/g
SG-Ref0 ^[68]^	none	n/a	0.12	116	5
SG-Ref0-(80)	none	n/a	0.34	151; 332	n.d.
SG-Ref100 ^[68]^	PEO	n/a	0.52	135; 1345	6
SG-CA91 ^[68]^	citric acid	7.5	1.14	140; 5320	5
SG-CA68 ^[68]^	citric acid	10	1.19	157; 3730	6
SG-CA45 ^[68]^	citric acid	15	1.04	137; 2000	7
SG-CA34 ^[68]^	citric acid	20	0.79	916	5
SG-CA27 ^[68]^	citric acid	25	0.67	326	9
SG-CA23 ^[68]^	citric acid	30	0.41	197	10
SG-CA19 ^[68]^	citric acid	35	0.25	164	12
SG-CA14 ^[68]^	citric acid	50	0.17	150	6
SG-C2#1 ^[91]^	oxalic acid	10	1.18	838	3
SG-C2#2 ^[91]^	oxalic acid	5	1.53	147; 2530	5
SG-C3 ^[91]^	malonic acid	10	0.75	259	10
SG-C3-(80)	malonic acid	10	1.39	786	n.d.
SG-C4 ^[91]^	succinic acid	10	0.43	182	8
SG-C5 ^[91]^	glutaric acid	10	0.38	195	8
SG-C6 ^[91]^	adipic acid	10	0.18	138	3

[a] ratio Al^3+^/ additive[b] calculated from mercury intrusion[c] calculated from nitrogen sorptionn.d. not determined

**Table 5 materials-13-01787-t005:** Pore diameters of AAO membranes as determined by SEM imaging.

Sample Name	Oxalic Acid AAO Treatment	Pore Size/nm (SEM)
AAO-0	(A) pristine	32–49
AAO-1100	(B) 1100 °C	29–47
AAO-Mn-900	(C) 900 °C, Mn-impregnated	42–46

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
