# Peer review of "Towards Macroporous α-Al2O3—Routes, Possibilities and Limitations"

_materials, 2020, doi:10.3390/ma13071787_

Round 1

Reviewer 1 Report

The manuscript is a combination of literature review and own results regarding fabrication of α-Al2O3 with high specific surface areas. In the own results part are too many reference to other articles, in terms of methods, compositions, results and interpretations (especially own [63] and [86]) and unpublished data. Also, are some conclusions where the authors mention the thermodynamic data without any reference.

All of these make the present manuscript to look more like a "review" than an "article".

There are some necessary revisions:

1. At pages 9, lines 368 and 389: Which was the heating rate from initial temperature to 1600°C? For calcination of a gel could be an important parameter.

2. At page 10, line 427: How were synthesized the granules? (as in section 3.1.1?) How much was their diameter?

3. At pages 22-24, Figures 14-16: For better view, I suggest to the authors to change rose dotted lines with other color (e.g. blue).

Author Response

Point 0

Yes, this manuscript may appear much like a review. Since the authors wishes to include their own results rather than split this manuscript into a review and an original research article, Materials advised us to submit it as an article (containing an extensive literature review section).
References to previous and future articles (now designated [68], [91], and [168]) have been reduced, and details have been added for several points.

Moreover, references [2 - 4] and [16] have been added to substantiate that α-Al2O3 is in fact the thermodynamically stable alumina modification under most conditions (p < 400 GPa, ABET < 130 m2/g).

Point 1

The heating rate for gel calcination to α-Al2O3 was 3 K/min. This information is in fact an important one for alumina gels and has now been added (line 368), thank you for pointing this out.

Point 2

Correct, sol-gel granules (2 - 5 mm) were synthesized as described in section 3.1.1. This information was missing and has now been added.

Point 3

We gladly took in the advice of changing the Hausmannite reflex indicators to a better legible light blue.

Reviewer 2 Report

This paper presents new results along with a review. It is a well-written and organized paper. Some comments are given below.

  1. What are the major mechanisms causing a considerable kinetic barrier for the conversion of transition alumina into corundum?
  2. The pore size and specific surface area of samples are reported in the paper. What is the pore volume of these samples? What is the desired porosity for the product?
  3. The effects of dopant ions and processing temperature on the specific surface area are significant. A comparison between Mn-ions and other dopants will be useful. 
  4. Is the temperature dependence of Mn-impregnation on ABET significant? If the temperature higher or lower 900 C, what is the change in ABET?

Author Response

Thank you for your supportive comments.

Point 1

The major kinetic barriers are found in the crystallite size incongruence, which has been emphasized further in the revised document, and in the considerably lower surface energy of γ-Al2O3s = 1.5 - 1.7 J/m2) compared to α-Al2O3s = 2.64 J/m2) [16]. The implications thereof can be seen when the decrease of specific surface, which is necessary for transition alumina to convert into corundum, is inhibited e.g., by pore-blocking effects.

Point 2

For all samples characterized by mercury intrusion, we have provided both the modal pore diameter(s) and the corresponding pore volume.
The targeted porosity is simply the highest possible one, while providing an α-Al2O3 stable above 1000 °C.

Point 3

Yes, it would indeed be interesting to thoroughly investigate other dopants, apart from the intensively discussed Mn and briefly studied Fe. However, we have based our choice on the comprehensive literature research presented in the review part and do believe to have presented results on the two most pertinent dopants in terms of facilitating the θ → α-transition. As mentioned, rare earths or other dopants often yield increased specific surface areas by preserving the spinel-type γ-Al2O3 structure.

Point 4

To address the temperature dependence of ABET in Mn-doped α-Al2O3, we have now included values of the Mn-free reference samples, calcined at 900 and 1050 °C, respectively, along with some discussion thereof. Unfortunately, due to the current situation in German University laboratories (SARS-CoV-2 pandemic), we are not able to provide further data on this subject at this point. However, studies are in preparation for a more detailed investigation of both the calcination temperature and duration for Mn-assisted sol-gel α-Al2O3. In case of novel insights and findings, a follow-up publication will be released.

Reviewer 3 Report

The authors present a very comprehensive review on the development of porous alpha-Al2O3. The report is very well written, and would surely serve as a key reference for those in the field.

Just a minor point, perhaps the authors would like to address;  the authors mention that obtaining a porous alpha-Al2O3 is largely due to thermodynamic constraints, yet, there is very little reference to thermodynanic studies showing this.

Author Response

Thank you for your supportive comments!
We gladly took in the advice of adding a few references ([2 - 4] and [16]) to substantiate that α-Al2O3 is in fact the thermodynamically stable alumina modification under most conditions (p < 400 GPa, ABET < 130 m2/g).

Round 2

Reviewer 1 Report

I agree with the publication of the manuscript in the present form.